# VITA: Vision-to-Action Flow Matching Policy

**Dechen Gao[1], Boqi Zhao[1], Andrew Lee[1], Ian Chuang[2], Hanchu Zhou[1], Hang Wang[1], Zhe Zhao[1], Junshan Zhang[1], Iman Soltani[1]**

[1]University of California, Davis, [2]University of California, Berkeley

{dcgao, boqzhao, awclee, hczhou, whang}@ucdavis.edu
{zao, jazh, isoltani}@ucdavis.edu
ianc@berkeley.edu

## ABSTRACT

Conventional flow matching and diffusion-based policies sample via iterative denoising from standard noise distributions (e.g., Gaussian), and require conditioning modules to repeatedly incorporate visual information during the generative process, incurring substantial time and memory overhead. To reduce the complexity, we develop VITA (**VI**sion-**T**o-**A**ction policy), a *noise-free* and *conditioning-free* flow matching policy learning framework that directly flows from visual representations to latent actions. Since the source of the flow is visually grounded, VITA eliminates the need for visual conditioning during generation. As expected, bridging vision and action is challenging, because actions are lower-dimensional, less structured, and sparser than visual representations; moreover, flow matching requires the source and target to have the same dimensionality. To overcome this, we introduce an action autoencoder that maps raw actions into a structured latent space aligned with visual latents, trained jointly with flow matching. To further prevent latent action space collapse during end-to-end training, we propose flow latent decoding, which anchors the latent generation process by backpropagating the action reconstruction loss through the flow matching ODE (ordinary differential equation) solving steps. We evaluate VITA on 9 simulation and 5 real-world tasks from ALOHA and Robomimic. VITA achieves $1.5\times$-$2\times$ faster inference compared to conventional methods with conditioning modules, while outperforming or matching state-of-the-art policies. Project page: VITA.

## 1 INTRODUCTION

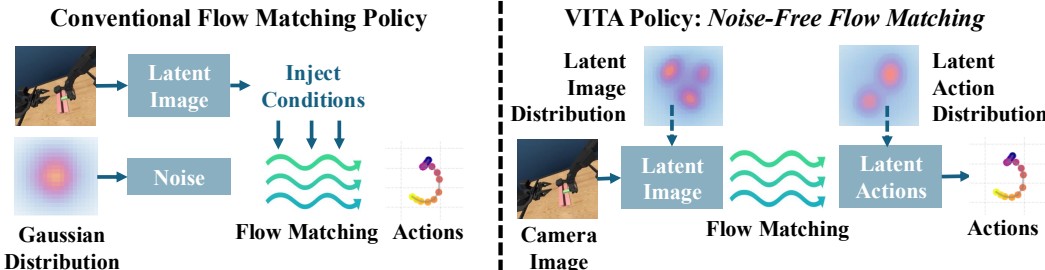

Figure 1: A comparison between VITA and conventional flow matching and diffusion policies. Unlike conventional methods that sample noise from standard distributions and inject input modalities via conditioning, VITA poses no constraints on the source distribution, and flows directly from latent visual representations to latent actions, eliminating the need for conditioning modules.

Flow matching and diffusion models have demonstrated remarkable success across a wide range of cross-modal generation tasks, from text-to-image generation (Rombach et al., 2022; Peebles & Xie, 2023; Ma et al., 2024; Liu et al., 2024a; He et al., 2025; Zhang et al., 2023), text-to-video generation (Ho et al., 2022; Li et al., 2023; Jin et al., 2024), to visuomotor (vision-to-action) policies (Chi et al., 2023; Ren et al., 2024a; Gao et al., 2025; Su et al., 2025; Zhang et al., 2025; Rouxel et al., 2024; Braun et al., 2024; Black et al.). Conventional flow matching and diffusion

methods (Lipman et al., 2023; Sohl-Dickstein et al., 2015) generate samples by starting with noise sampled from a basic source distribution (often Gaussian) and progressively "denoising" them into the target modality. This process requires repeatedly injecting visual information at each denoising step through additional conditioning modules (Rombach et al., 2022; Zhang et al., 2023; Chi et al., 2023; Dasari et al., 2024), resulting in substantial time and space overheads (Liu et al., 2024a; He et al., 2025). In particular, cross-attention, AdaLN (Peebles & Xie, 2023), or FiLM (Perez et al., 2018) are often used for visual conditioning. Cross-attention incurs quadratic time and space complexity, while AdaLN and FiLM avoid quadratic complexity but require extra modulation networks to generate feature-wise parameters at every denoising step.

Minimizing complexity is essential for real-time robot control, e.g., Pi-0.5 (Intelligence et al.) operates at 50 Hz and Helix (Figure AI, 2025) at up to 200 Hz, imposing stringent requirements on inference latency. The primary objective of this paper is to overcome the inefficiencies inherent to conditioning mechanisms in conventional flow matching methods. To this end, we develop VITA (**VI**sion-**T**o-**A**ction policy), a noise-free flow matching policy learning framework that directly maps visual representations to latent actions. As depicted in Figure 1, unlike conventional methods that flow from a Gaussian prior, VITA imposes no constraints on the source distribution and flows directly from visual latents, obviating the need for repeated visual conditioning during the flow. Consequently, VITA significantly reduces time and space overheads, and simplifies network architectures.

Learning vision-to-action flow matching, however, presents several new challenges. Bridging two distinct modalities is inherently difficult (Liu et al., 2024a), particularly in robotics where action data is limited, unstructured, and sparse, whereas visual representations exhibit rich structures and semantics, and far higher dimensions. Additionally, flow matching requires that the source and target have equal dimensionalities, which prevents directly aligning raw actions with visual representations.

To address these challenges, we propose two key designs for VITA. 1) **Learning Latent Actions as Flow Matching Targets.** We introduce a latent action space, learned via an action autoencoder, that 'lifts' action representations to match the higher dimensionality of visual representations and serves as a structured target distribution for flow matching. The action encoder up-samples raw actions into target latent actions, and a decoder reconstructs raw actions from these latents. 2) **Enabling End-to-End Latent Action Policy Learning via Flow Latent Decoding.** In conventional flow matching methods such as latent diffusion for image generation (Rombach et al., 2022), the target latent space can be pre-trained with abundant images and then frozen as reliable flow matching targets; in contrast, we show that a pre-trained and frozen latent action space for flow matching yields poor performance (discussed in Appendix B.2), since action data is too sparse and limited to learn reliable targets and frozen targets cannot be corrected. It is therefore necessary to jointly train the flow model with the action autoencoder. To prevent collapse of the latent action space (which happens by naively reducing flow matching and autoencoder losses), we introduce flow latent decoding which backpropagates the reconstruction loss of latent actions generated by solving flow matching ordinary differential equations (ODEs), anchoring latent generation using ground-truth actions. This approach bridges the *training-inference gap of latent actions*: during training, the flow matching model learns to match the targets given by the action encoder, and the decoder is trained to reconstruct these targets, whereas at test time, the decoder must reconstruct ODE-generated latent actions.

We evaluate VITA on both real-world and simulated tasks using ALOHA (Chuang et al., 2024; Fu et al., 2024b; Zhao et al., 2024) and Robomimic (Mandlekar et al., 2021). VITA achieves $1.5\times$-$2\times$ faster inference and 18.6%-28.7% lower memory usage compared to conventional flow matching with similar model sizes, while outperforming or matching state-of-the-art policies in success rates. Additionally, compared to state-of-the-art methods that *necessitate* complex architectures like transformers (Ma et al., 2024), VITA naturally simplifies architecture designs. For instance, with vector-based visual representations, VITA reduces the flow-matching network to a conditioning-free vector-to-vector mapping, allowing for the use of simple MLPs; with higher-dimensional grid-based visual representations, VITA scales to more complex architectures such as transformers while eliminating costly conditioning modules like cross-attention. We further demonstrate that VITA achieves stable learning with fast convergence while maintaining high precision (discussed in Appendix B.8).

Our main contributions are summarized as follows:

**Noise-Free Flow Matching for Visuomotor Learning.** We propose VITA, a noise-free policy that directly evolves latent visual representations into latent actions via flow matching. VITA learns a

structured latent action space aligned with visual representations to bridge the modality gap. To prevent latent action collapse during end-to-end latent action policy training, we propose flow latent decoding, which refines latent actions by backpropagating through the flow matching ODE steps.

**Efficient Policy Architectures.** By visually grounding the source of the flow, VITA obviates costly conditioning required by flow matching policies to repeatedly inject visual inputs. VITA enables lightweight implementations. To our knowledge, VITA is the first MLP-only flow matching policy to succeed on tasks as challenging as ALOHA bimanual manipulation.

**State-of-the-Art Efficiency and Performance.** We validate VITA's efficiency and performance on 9 simulated and 5 real-world tasks spanning both bimanual and single-arm manipulation. VITA delivers $1.5\times$-$2\times$ faster inference and 18.6%-28.7% lower memory usage compared to conventional flow matching, while surpassing or matching state-of-the-art policies in success rates.

## 2 RELATED WORK

**Imitation Learning for Visuomotor Policy.** Imitation learning enables robots to learn complex behaviors by mimicking expert demonstrations. Behavior cloning is a prominent imitation learning paradigm that learns a policy mapping observations to actions via supervised learning (Zhao et al., 2023; Fu et al., 2024a; Gong et al., 2024; Su et al., 2025). Recent advancements in behavioral cloning increasingly leverage generative modeling (Chi et al., 2023), which models the conditional action distribution given observations using conditional variational autoencoders (CVAEs) (Zhao et al., 2023; Lee et al., 2024), diffusion models (Dasari et al., 2024; Chi et al., 2023), or flow matching (Zhang & Gienger, 2024; Zhang et al., 2025). These approaches commonly rely on conditioning mechanisms (e.g., cross-attention (Dasari et al., 2024), AdaLN (Dasari et al., 2024), FiLM (Perez et al., 2018)) to incorporate visual observations during the generative process. In contrast, VITA removes the need for visual conditioning modules by developing a noise-free vision-to-action flow.

**Diffusion and Flow Matching for Generative Modeling.** Unlike diffusion, which samples from Gaussian distributions, flow matching theoretically places no constraints on the choice of source distribution (Tong et al., 2024). A few works have explored this property to learn the direct transport within the same modality (Albergo & Vanden-Eijnden, 2022; Tong et al., 2023b). Recently, Liu et al. (2024a) and He et al. (2025) extended this to more challenging cross-modal generation between text and image. VITA learns to bridge vision and action for visuomotor control, where the action modality presents unique challenges because of limited data and its unstructured nature. Different from flow matching for image generation, which typically pre-trains and freezes the latent image space when learning the flow (Rombach et al., 2022; Liu et al., 2024a), VITA resorts to a fully end-to-end pipeline training to effectively learn the latent action space from limited and sparse action data along with flow matching. We propose flow latent decoding to backpropagate action reconstruction losses through the latent action generation process (ODE solving steps) during training.

## 3 PRELIMINARIES

Flow matching models learn to transport samples from a source distribution $p_0$ to a target distribution $p_1$ by learning a velocity vector field $v_\theta$. The generative process is defined by an ODE $\frac{d\mathbf{z}_t}{dt} = v_\theta(\mathbf{z}_t, t)$ (Liu et al., 2022c; Wang & Qureshi, 2023), where $t \in [0, 1]$ is continuous time, and $\mathbf{z}_t$ denotes a sample at time $t$. The goal is for the learned flow to transport $\mathbf{z}_0 \sim p_0$ to $\mathbf{z}_1 \sim p_1$.

**Training.** For a linear interpolation between two samples, the interpolation path is $\mathbf{z}_t = (1-t)\mathbf{z}_0 + t\mathbf{z}_1$. The ground-truth velocity along this path is $\frac{d\mathbf{z}_t}{dt} = \mathbf{z}_1 - \mathbf{z}_0$. The flow matching loss trains $v_\theta$ to match this supervised vector field:

$$\mathcal{L}_{\text{FM}} = \mathbb{E}_{t, \mathbf{z}_0, \mathbf{z}_1} \left[ \|v_\theta(\mathbf{z}_t, t) - (\mathbf{z}_1 - \mathbf{z}_0)\|^2 \right]. \tag{1}$$

**Inference.** Given a source sample $\mathbf{z}_0$, a target sample is obtained by solving the ODE from $t = 0$ to $t = 1$: $\hat{\mathbf{z}}_1 = \mathbf{z}_0 + \int_0^1 v_\theta(\mathbf{z}_t, t)\, dt$. In practice, we apply an Euler solver with $K$ discretization steps, yielding updates of the form $\mathbf{z}_{t_{k+1}} = \mathbf{z}_{t_k} + \Delta t\, v_\theta(\mathbf{z}_{t_k}, t_k)$, where $\Delta t = 1/K$.

# 4 VITA: VISION-TO-ACTION FLOW MATCHING

The key challenge in VITA is the large dimensionality gap between vision and action, compounded by the sparsity and unstructured nature of action data. In this section, we present the core designs of VITA developed to address these issues. We first introduce the mathematical formulation of VITA (Section 4.1) and its overall architecture (Section 4.2). We then show why constructing a latent action space is essential for resolving dimensionality mismatch (Section 4.3), and propose flow latent decoding to address latent action collapse. Finally, we describe the objectives that enable effective end-to-end VITA learning from scratch (Section 4.4).

## 4.1 FLOWING FROM VISION TO ACTION

VITA learns a policy $\pi(A|O)$ that directly maps observations $O$ to a corresponding sequence of future actions $A$. The observations $O$ encompass raw visual inputs $I \in \mathbb{R}^{H \times W \times C}$ and, optionally, the robot's proprioceptive states $S$. Actions are represented as temporal sequences over a prediction horizon, formally defined as $A \in \mathbb{R}^{T_{\text{pred}} \times D_{\text{action}}}$, where $T_{\text{pred}}$ is the prediction horizon and $D_{\text{action}}$ is the dimensionality of the action space. We employ action chunking with $T_{\text{pred}} > 1$ to enhance temporal consistency (Zhao et al., 2023).

**Conventional vs. VITA Flow Matching.** Conventional flow matching policies generate actions by evolving samples from a noise prior, typically $\mathbf{z}_0 \sim \mathcal{N}(0, \mathcal{I})$. To incorporate visual information, these models learn a conditional velocity field $v_\theta(\mathbf{z}_t, t \mid O)$, requiring conditioning modules (e.g., cross-attention) to inject observations $O$ at every denoising step. In contrast, VITA directly treats the visual latent as the source of the flow $\mathbf{z}_0$. Because the flow is visually grounded at the source, VITA learns a conditioning-free velocity field for flow matching, $v_\theta(\mathbf{z}_t, t)$, eliminating the need for repetitive conditioning and yielding a noise-free framework with enhanced efficiency.

Critically, flow matching requires $\mathbf{z}_0$ and $\mathbf{z}_1$ to share identical dimensionality, necessitating the construction of a latent action space that matches the dimensionality of visual representations (Section 4.3). During inference, the current observation $O_{\text{curr}}$ is first encoded into its latent visual representation $\mathbf{z}_0 = \mathcal{E}_v(O_{\text{curr}})$, which is subsequently evolved into a predicted latent action representation, $\hat{\mathbf{z}}_1$, by numerically solving the ODE from $t = 0$ to $t = 1$ using the learned velocity field $v_\theta$. In other words, $\hat{\mathbf{z}}_1$ is an approximation to the target latent $\mathbf{z}_1$. The resulting latent action $\hat{\mathbf{z}}_1$ is then decoded through the action decoder to yield the final action sequence $\hat{A} = \mathcal{D}_a(\hat{\mathbf{z}}_1)$.

## 4.2 VITA ARCHITECTURE DESIGN

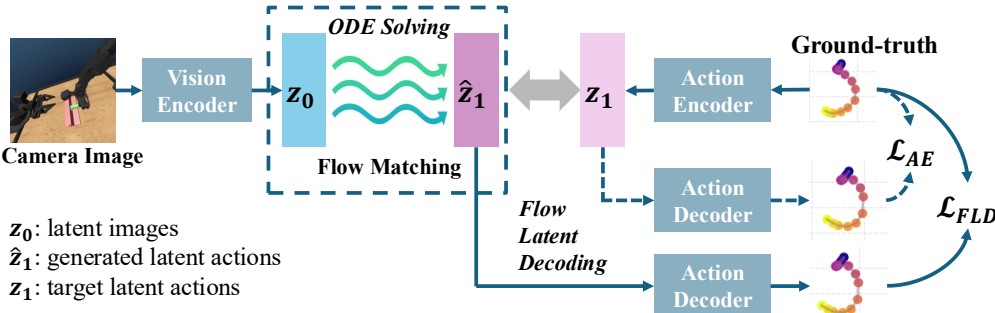

$\mathbf{z}_0$: latent images
$\hat{\mathbf{z}}_1$: generated latent actions
$\mathbf{z}_1$: target latent actions

Figure 2: An overview of the VITA architecture: The vision encoder maps observations into a source latent representation $\mathbf{z}_0$ for the flow; the action encoder provides a target latent representation $\mathbf{z}_1$ for flow matching training. The action decoder learns to decode $\hat{\mathbf{z}}_1$ (latent actions generated by solving ODEs) to actions via flow latent decoding losses, and decode $\mathbf{z}_1$ to actions (latent actions from action encoder) via autoencoder losses. The flow matching network learns the velocity field over a continuous flow matching path from $\mathbf{z}_0$ to $\mathbf{z}_1$.

As depicted in Figure 2, VITA is composed of three primary components: 1) The **Vision Encoder** ($\mathcal{E}_v$) maps observations $O$ to the latent representation $\mathbf{z}_0 = \mathcal{E}_v(O)$, where $\mathbf{z}_0 \in \mathbb{R}^{D_{\text{latent}}}$ serves as the source of the flow. 2) The **Action Autoencoder (AE)** consists of the Action Encoder and the Action Decoder, and learns latent representations for action chunks. The Action Encoder ($\mathcal{E}_a$) maps the

ground-truth action chunk $A$ to latent actions $\boldsymbol{z}_1 = \mathcal{E}_a(A)$, where $\boldsymbol{z}_1 \in \mathbb{R}^{D_{\text{latent}}}$ serves as the target for flow matching; the Action Decoder ($\mathcal{D}_a$) reconstructs an action chunk $\hat{A} = \mathcal{D}_a(\hat{\boldsymbol{z}}_1)$ from latent actions $\hat{\boldsymbol{z}}_1$. 3) The **Flow Matching Network** ($v_\theta$) is learned to predict the velocity field at arbitrary $t$.

## 4.3 BRIDGING THE MODALITY GAP BETWEEN VISION AND ACTION

A key constraint of flow matching is that the source and target distributions must share the same dimensionality. This poses a critical challenge for vision-to-action policies, since action spaces are typically much lower-dimensional than visual representations. For example, action dimensionalities range from 2 on `PushT` to 21 on `ThreadNeedle`, whereas visual representations can be 512-dimensional (Zhao et al., 2024) or even higher when using grid-based features.

To bridge the gap, one naive option is to down-sample latent visual representations to the action chunk dimensionality, which, however, causes severe information loss and degrades performance. Alternatively, one can up-sample actions with zero-padding, yielding sparse and unstructured targets that hinder flow matching training (see Appendix B.1). A third alternative is a pre-trained, frozen action AE, akin to common practice in latent diffusion for image generation (Rombach et al., 2022), but this proves ineffective for learning flow matching over latent actions: with sparse and limited action data, the induced latent space is unreliable as a flow target and cannot be corrected once frozen (see Appendix B.2). As another alternative, jointly training the action AE with flow matching may still fail; our empirical studies identify the root cause as latent action space collapse induced by the training-inference gap in action decoder inputs, as detailed below.

**Training-Inference Gap between Encoder-Based and ODE-Generated Latent Actions.** During training, the decoder reconstructs actions from encoder-based latent actions $\boldsymbol{z}_1$, whereas at inference it decodes $\hat{\boldsymbol{z}}_1$ generated by solving the flow matching ODE. Since $\hat{\boldsymbol{z}}_1$ is an approximation and does not always align with $\boldsymbol{z}_1$, the decoder can fail to map them into meaningful actions. To address this gap, we propose flow latent decoding, which enforces the model to decode from ODE-generated latent actions $\hat{\boldsymbol{z}}_1$ during training, anchoring the latent generation process with ground-truth actions.

## 4.4 VITA LEARNING OBJECTIVES

Building upon our analysis of the training-inference gap, we now formulate a comprehensive learning framework for VITA that prevents latent collapse and ensures effective end-to-end optimization. Our framework includes three essential objectives: flow latent decoding (FLD), flow matching (FM), and action autoencoder (AE) losses, each addressing distinct aspects of the learning challenge.

**Flow Latent Decoding (FLD).** FLD addresses the training-inference gap by anchoring ODE-generated actions using ground-truth actions during training. Formally, FLD is defined as the reconstruction loss using ODE-generated latent actions, $\mathcal{L}_{\text{FLD}} = \left\| \mathcal{D}_a(\hat{\boldsymbol{z}}_1) - A \right\|$, where $\hat{\boldsymbol{z}}_1$ is obtained by solving the flow ODE with an Euler solver during training. FLD propagates gradients through the decoder and the ODE solver into both $v_\theta$ and $\mathcal{E}_v$. By decoding $\hat{\boldsymbol{z}}_1$ into actions and measuring reconstruction error directly in action space, FLD effectively minimizes the discrepancies between encoder-based and ODE-generated latents.

**Flow Latent Consistency (FLC).** To gain deeper insight into the mechanics of FLD, we introduce flow latent consistency (FLC), a minimalist surrogate that directly aligns ODE-generated and encoder-based latents without decoding. Formally, FLC is defined as $\mathcal{L}_{\text{FLC}} = \left\| \hat{\boldsymbol{z}}_1 - \boldsymbol{z}_1 \right\|$. Under mild local regularity assumptions on $\mathcal{D}_a$ (stated below), FLC and FLD provide locally equivalent training signals for the same $\hat{\boldsymbol{z}}_1$. Empirically, FLC also prevents collapse without explicit action reconstruction, though convergence is slightly slower than with FLD (Section 5.3). This theoretical connection not only validates our approach but also offers computational flexibility in implementation. A sketch of the analysis is given below, with full proofs and corollaries deferred to Appendix A.

**Assumption 1** (Decoder locally well-behaved). *$\mathcal{D}_a$ is $C^1$ in a neighborhood of $\boldsymbol{z}_1$, with Jacobian singular values bounded as $m \le \sigma_{\min} \le \sigma_{\max} \le L$ on that neighborhood. Let $\varepsilon_{\text{AE}} := \left\| \mathcal{D}_a(\boldsymbol{z}_1) - A \right\|$ denote the local AE reconstruction error.*

**Theorem 1** (Local equivalence of FLD and FLC). *Under Assumption 1, for any $\hat{\boldsymbol{z}}_1$ in the neighborhood, we have $m \left\| \hat{\boldsymbol{z}}_1 - \boldsymbol{z}_1 \right\| - \varepsilon_{\text{AE}} \le \left\| \mathcal{D}_a(\hat{\boldsymbol{z}}_1) - A \right\| \le L \left\| \hat{\boldsymbol{z}}_1 - \boldsymbol{z}_1 \right\| + \varepsilon_{\text{AE}}.$*

*If $\varepsilon_{\text{AE}} = 0$, the minimizers of $\mathcal{L}_{\text{FLD}}$ and $\mathcal{L}_{\text{FLC}}$ coincide and equal $\{z_1\}$; if $\varepsilon_{\text{AE}} > 0$, any minimizer of $\mathcal{L}_{\text{FLD}}$ lies within radius $\varepsilon_{\text{AE}}/m$ of $z_1$.*

This theoretical result confirms that FLD and FLC target the same underlying optimization objective.

We further discuss this in Section 5.3, showing that including the flow latent decoding loss with a non-zero $\lambda_{\text{FLD}}$ is critical for avoiding latent space collapse and training successful VITA policies. We will also ablate the effects of $\lambda_{\text{FLD}}$, $\lambda_{\text{FLC}}$, and $\lambda_{\text{AE}}$ in Figure 6.

**Flow Matching and Autoencoder Losses.** The flow matching loss supervises the flow network $v_\theta$ by minimizing the MSE between the predicted velocity and the ground-truth velocity as shown in Equation (1). The action autoencoder loss trains $(\mathcal{E}_a, \mathcal{D}_a)$ to reconstruct action chunks using an L1 loss, $\mathcal{L}_{\text{AE}} = \|A - \mathcal{D}_a(\mathcal{E}_a(A))\|_1$, where $z_1 = \mathcal{E}_a(A)$ serves as a structured target latent with small reconstruction bias and good local conditioning. This structured latent space strengthens the theoretical link between FLD and FLC, and further stabilizes training.

The training objective is a weighted sum of all three losses: $\mathcal{L}_{\text{VITA}} = \lambda_{\text{FM}}\mathcal{L}_{\text{FM}} + \lambda_{\text{FLD}}\mathcal{L}_{\text{FLD}} + \lambda_{\text{AE}}\mathcal{L}_{\text{AE}}$.

# 5 EXPERIMENTS

We evaluate VITA on 9 simulation and 5 real-world tasks spanning both bimanual and single-arm manipulation. The bimanual tasks include 5 simulation (Figure 3) and 2 real-world tasks (Figure 4) on AV-ALOHA (Chuang et al., 2024), which augments ALOHA (Zhao et al., 2024) with an active-vision camera mounted on an additional 7-DoF arm. This challenging suite features high-precision requirements, non-stationary observations enabled by active vision, and 21-DoF high-dimensional actions. The single-arm tasks include 3 real-world tasks using one ALOHA arm (Figure 4), featuring high randomization in object positions or colors, and 4 simulated tasks, including 2 Robomimic tasks (7D actions), PushT (2D actions), and CloseBox (9D actions) from RLBench (James et al., 2020).

For simulated tasks, each environment provides 100-200 demonstrations. AV-ALOHA demonstrations were collected via expert teleoperation in VR, using the left-eye image as input; single-arm ALOHA demonstrations were collected using a leader arm. For the remaining tasks, we use publicly available datasets. In real-world AV-ALOHA experiments, each task is trained from 50 demonstrations using the left stereo image, and the single-arm ALOHA tasks are trained from 50-100 demonstrations using the wrist camera and an overhead camera. See Appendix D for details.

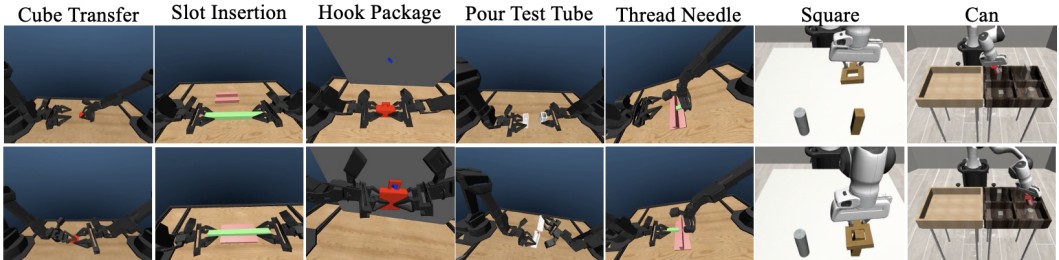

Figure 3: Autonomous rollouts of VITA across 5 AV-ALOHA tasks (CubeTransfer, SlotInsertion, HookPackage, PourTestTube, ThreadNeedle), and 2 Robomimic tasks (Square, Can). Notably, the AV-ALOHA tasks demand high-precision control, such as accurately pouring a small ball into a narrow tube opening, or threading a needle through a tiny hole.

## 5.1 EXPERIMENT SETTINGS

**Flow Matcher.** We adopt OT-CFM (Tong et al., 2023a), which is based on optimal transport, and solve the ODE with an Euler solver using 6 linearly interpolated time steps $t$ for both VITA and FM.

**Vision Encoding.** We use ResNet-18 (He et al., 2016) as the vision encoder. A common practice in visuomotor policy learning is to use the vector-based visual features after global average pooling (Su et al., 2025; Chi et al., 2023), in which case VITA operates entirely on vector representations for both

| Hidden Pickup | Transfer From Box | Pick Ball | Tooth Brush | Store Drawer |

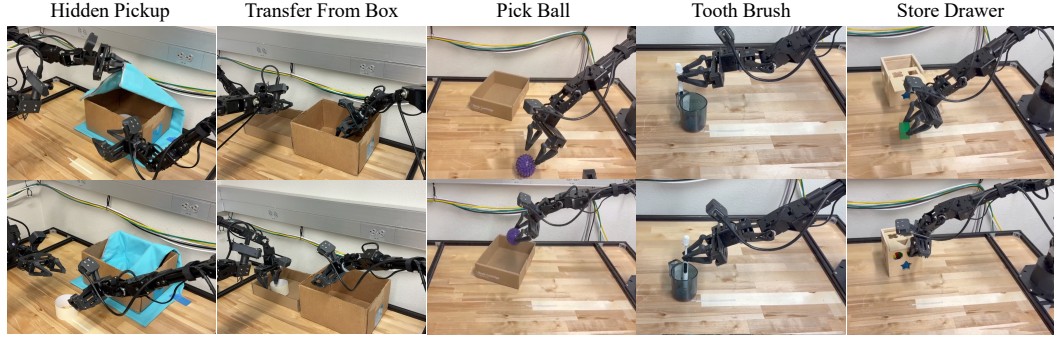

Figure 4: Autonomous rollouts of VITA on five challenging real-world tasks, including two bimanual AV-ALOHA tasks, `HiddenPick`, and `TransferFromBox` using active vision, and three single-arm ALOHA tasks, `PickBall`, `ToothBrush`, and `StoreDrawer`

vision and action. We additionally evaluate a variant of VITA using the $9 \times 512$ ResNet grid-based features to preserve more spatial information and assess VITA's scalability.

**Baselines.** Our baselines include state-of-the-art policies, including flow matching (FM) policy (Zhang & Gienger, 2024), diffusion policy (DP) (Chi et al., 2023), and action chunking transformer (ACT) (Zhao et al., 2023). We evaluate both efficiency and performance in Section 5.2.

**Training.** We train VITA and baselines to predict action chunks of length 16, of which the first 8 actions are executed. We train VITA and FM on each task for 25K-50K steps. We follow the ACT and DP implementations in LeRobot (Cadene et al., 2024). Since VITA and FM converge much faster than DP or ACT (a known advantage of FM methods (Lipman et al., 2023)), we extend DP training to 100K steps, and ACT to 100K-200K steps. The training batch size is 128. See Appendix G for more detailed training settings.

**Evaluation.** We use 6 ODE steps for VITA and FM, and 10 DDPM (Denoising Diffusion Probabilistic Model) steps for DP (Ho et al., 2020). In simulation, we evaluate the policy every 200 training steps using 50 episodes per evaluation, and report the best success rate (SR) averaged over 3 seeds (Table 2). For real-world tasks, we evaluate over 20 episodes per checkpoint on three single-arm tasks (Table 3) and two bimanual manipulation tasks (Appendix D.2.1). Efficiency gains of VITA are analyzed in Section 5.2.1, with all latency and memory measurements obtained on a single NVIDIA RTX 4090.

## 5.2 PERFORMANCE

The main objective of this paper is to improve the efficiency of visuomotor policies. This section demonstrates that VITA is a fast and precise visuomotor control policy, delivering efficiency gains (Section 5.2.1) while matching or surpassing state-of-the-art methods in success rates (Section 5.2.2). Appendix B.8.2 further analyzes VITA's advantages in fast convergence and high precision. Appendix B.8 demonstrates VITA's robustness to online perturbations, and Appendix E.2 evaluates its generalization to unseen objects.

### 5.2.1 EFFICIENCY

FM and DP are typically parameterized using U-Nets (Ronneberger et al., 2015) or diffusion transformers (DiTs) (Peebles & Xie, 2023), which predict velocity fields or noise at each denoising step. U-Nets and DiTs are often large and computationally costly, which limits their real-time deployment that requires highly efficient inference. These architectures rely on explicit visual conditioning modules, e.g., cross-attention, FiLM, or AdaLN, which must be executed at every denoising step and inevitably increase both inference time and memory footprint.

A key determinant of efficiency in both conventional FM and VITA is the choice of visual latent representation. In conventional methods, vector-based and grid-based latents require different conditioning modules and thus incur different computational costs. In VITA, the visual and action latents must share the same dimensionality, so the latent choice directly determines the FM network

architecture. We analyze the implementations and efficiency of VITA and conventional FM in these two settings and explain why VITA yields efficiency advantages in both.

**Vector-Based Visual Latents.** With vector-based visual features, conventional methods typically employ FiLM (Perez et al., 2018) or AdaLN (Peebles & Xie, 2023) to condition the FM network. These methods compute modulation parameters via a separate conditioning network at every denoising step. FiLM modulates network outputs at each feature channel; AdaLN modulates normalization statistics at each network layer. The FM network is often implemented using transformers or U-Nets to effectively process noisy action chunks of shape $T_{\text{pred}} \times D_{\text{action}}$, and fuse in visual latents. In contrast, since VITA uses vector-based latents for both the vision source and action targets, the FM network $v_\theta$ reduces to a vector-to-vector mapping, with no need to fuse visual information. This enables a highly lightweight MLP-only architecture choice. We show that MLP-based VITA achieves better efficiency (both latency and memory) than the most efficient FM baseline (MLP-based), while matching the task performance of the strongest FM baseline (transformer-based).

**Grid-Based Visual Latents.** With grid-based features (e.g., $9 \times 512$), cross-attention is often used in transformers to fuse visual tokens and action chunks for conditioning (Dasari et al., 2024), which introduces quadratic time and space complexity. In contrast, VITA eliminates cross-attention in transformer-based implementations. As shown in Appendix B.6.2, VITA attains strong performance while being more efficient (Table 1 grid-based settings).

To provide a comprehensive efficiency comparison, we implement two FM parameterizations (DiTs (Peebles & Xie, 2023) and U-Nets (Ronneberger et al., 2015)) and three conditioning mechanisms: AdaLN (Peebles & Xie, 2023) and FiLM (Perez et al., 2018) for vector-based features, and cross-attention (Gong et al., 2024) for grid-based features.

We compare the inference latency and inference memory usage of VITA and other FM methods in Table 1 for both vector-based and grid-based representations. VITA achieves an inference wall-clock time of 0.22 ms (vector-based) and 0.25 ms (grid-based) per action chunk, which is $1.5\times$ and $2\times$ faster than the best-performing FM baseline (transformer-based FM with similar model sizes) that incurs higher latency ($\sim$0.33 ms and $\sim$0.51 ms). VITA effectively reduces memory usage: the peak memory is 18.6% less than FM in the vector-based setting, and 28.7% less in the grid-based setting. Additionally, we discuss the training time and space efficiency in Appendix B.7. We further compare against FM accelerated with simple MLP architectures of similar model size to isolate the effect of architecture design in the vector-based setting. VITA remains $1.3\times$ faster at inference and uses 19.3% less memory, while FM with MLPs fails to achieve competitive success rates (Appendix B.6.1).

Table 1: Comparison of the time and space efficiency of VITA and flow-matching baselines, grouped by the type of visual latents used ("Vector" or "Grid" based). Metrics include: model size, inference latency (ms/chunk, batch size 1), and inference memory (MiB), (see Appendix B.7.2 for inference memory measurement details).

| Visual | Model | Architecture | Conditioning | Params | Latency | Memory |
|--------|-------|--------------|--------------|--------|---------|--------|
| **Vector** | VITA | MLP | *N/A* | 31.09M | **0.2215** | **333.86** |
| | FM | Transformer | AdaLN | 31.16M | 0.3307 | 410.38 |
| | FM | U-Net | FiLM | 84.05M | 0.3650 | 818.79 |
| | FM | MLP | AdaLN | 32.20M | 0.2831 | 413.95 |
| | DDPM | U-Net | FiLM | 81.82M | 2.5985 | 801.47 |
| **Grid** | VITA | Transformer | *N/A* | 31.80M | **0.2502** | **377.55** |
| | FM | Transformer | Cross-Attn | 29.06M | 0.5102 | 529.16 |

### 5.2.2 SUCCESS RATES

We evaluate VITA against state-of-the-art policies on 9 simulated tasks and 5 real-world tasks. We report SRs for FM using the transformer-based FM with AdaLN, as it achieves the strongest performance among the FM variants evaluated in Section 5.2.1; other variants, such as MLP-based FM, perform poorly (Appendix B.6.1), while FM using cross-attention or U-Net reaches similar SRs but is substantially slower to train. Across the 9 simulated tasks (Table 2) and the 3 real-world single-arm ALOHA tasks (Table 3), VITA consistently outperforms or matches state-of-the-art methods. We further evaluate VITA on 2 challenging real-world bimanual AV-ALOHA tasks (Appendix D.2.1)

featuring 21-DoF high-dimensional actions and active vision. We discuss the under-performance of DP in Appendix B.8.2, which is largely due to the high-precision requirements and stringent success criteria of multi-stage ALOHA tasks.

Table 2: SRs on simulated tasks on AV-ALOHA, Robomimic, `PushT`, and `CloseBox`. We report the mean and the standard deviation of the best SRs during validation across 3 random seeds.

| Task | VITA | FM | DP | ACT |
|------|------|-----|-----|-----|
| ThreadNeedle | **91.33**±1.15 | 90±2 | 59.33±1.89 | 44.67±14.47 |
| SlotInsertion | 78±2 | **82**±2 | 50.67±5.03 | 47.33±2.31 |
| PourTestTube | 78.67±2.31 | **86**±2.31 | 46±0 | 42±7.21 |
| HookPackage | **86**±2 | 82±2 | 37.33±6.11 | 32±2 |
| CubeTransfer | **100**±0 | **100**±0 | 94.67±3.06 | 99.33±1.16 |
| PushT | **88**±2 | 83.33±1.16 | 74.67±6.11 | 28±5.29 |
| Square | **95.33**±4.16 | 87.33±3.06 | 84±2 | 72±2 |
| Can | **100**±0 | **100**±0 | 95.33±1.16 | 88.67±2.31 |
| CloseBox | **95.33**±1.16 | 85.33±2.31 | 85.33±1.16 | 72±5.29 |

Table 3: Comparison of SRs on three real-world single-arm ALOHA manipulation tasks. Each task is decomposed into subtasks, and SRs are reported per subtask.

| | PickBall | | StoreDrawer | | | ToothBrush | |
|------|------|------|------|------|------|------|------|
| | Pick | Place | Pick | Place | Close | Pick | Insert |
| **VITA** | **0.75** | **0.70** | **1.00** | **0.95** | **0.95** | 0.80 | **0.50** |
| **FM** | **0.75** | 0.65 | 0.90 | 0.90 | 0.90 | **0.90** | **0.50** |
| **DP** | 0.60 | 0.60 | **1.00** | **0.95** | 0.90 | 0.60 | 0.30 |
| **ACT** | 0.50 | 0.45 | 0.65 | 0.65 | 0.50 | 0.70 | 0.30 |

## 5.3 ABLATION OF FLOW LATENT DECODING

We investigate the importance of FLD for effectively training VITA policies. As demonstrated by Figure 5(a), the latent actions generated at inference may not decode into meaningful actions because of the training-inference gap between encoder-based latent actions $z_1$ and ODE-generated latent actions $\hat{z}_1$ discussed in Section 4.3. We propose two objectives that backpropagate through ODEs to minimize the discrepancy: 1) FLD, which enforces accurate reconstruction in the raw action space by comparing $\mathcal{D}_a(\hat{z}_1)$ against the ground-truth action; and 2) FLC, which enforces alignment in the latent action space between ODE-generated latents $\hat{z}_1$ and encoder-based latents $z_1$.

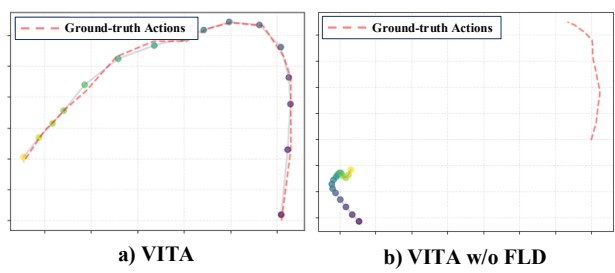

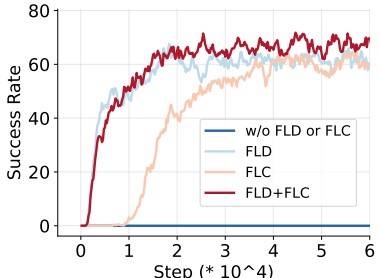

Figure 5: Comparison of reconstructed actions between (a) VITA, and (b) VITA without FLD. Reconstruction fails without FLD because of latent space collapse.

Figure 6: Success rates using different objectives.

Figure 6 shows that the model completely fails to learn without FLD (i.e., $\lambda_{\text{FLD}} = 0$) due to latent collapse, while applying FLD succeeds in learning the policy. FLC provides a weaker signal compared to FLD (because FLD directly anchors generation using the ground-truth actions), resulting in slightly

slower convergence. However, we found that using a combination of both objectives in practice yields the best performance due to richer learning signals in both raw and latent action spaces.

### 5.4 VITA DENOISING: FROM VISUAL LATENTS TO PRECISE ACTIONS

Figure 7 compares the denoising processes of VITA and conventional flow matching. In conventional methods, the ODE solver transports samples from an uninformative isotropic Gaussian prior to a highly structured action space, and therefore requires repeated visual conditioning to guide denoising. In contrast, VITA initializes the flow directly from visual representations. Owing to the end-to-end joint optimization enabled by FLD, the visual and action latent spaces are co-evolved, leading to **closely aligned latent manifolds of vision and action** in practice. Empirically, we observe that the latent visual representations already encode coarse action semantics and can be directly decoded into preliminary action trajectories even before any ODE integration, demonstrating that VITA learns an **action-centric visual representation**. The close alignment between vision and action manifolds also explains why a lightweight MLP suffices for VITA, whereas MLP-only FM struggles due to the need to transport from an unstructured Gaussian prior to a structured action space (detailed in Appendix B.6.1).

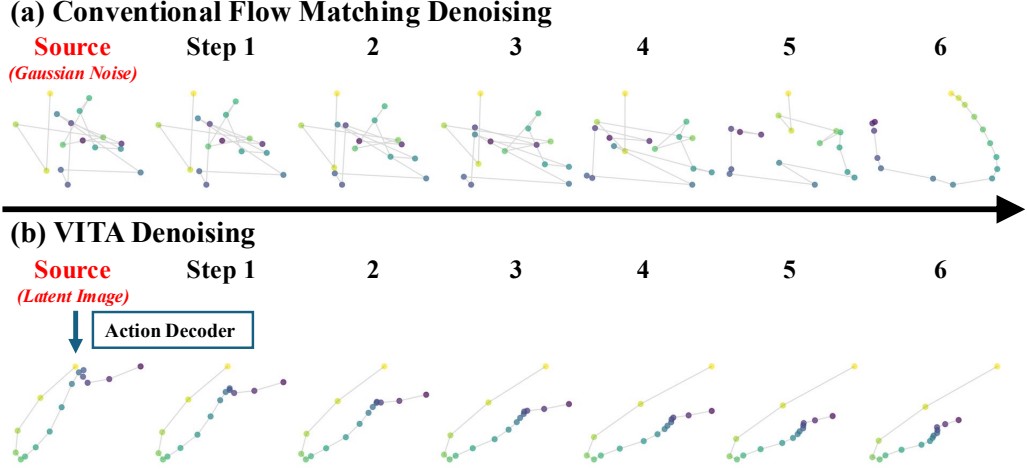

Figure 7: (a) Conventional flow matching transports Gaussian noise into action chunks, requiring repeated injection of visual information to shape action trajectories. (b) VITA initializes the flow from visual representations. Remarkably, the end-to-end optimization induces action-centric visual representations aligned with the latent action space. As a result, the initial visual latents can be directly decoded into preliminary action trajectories, which the ODE solver then smoothly refines into precise actions.

## 6 CONCLUSION

We developed VITA, an efficient and high-performing visuomotor policy that generates actions in a noise-free manner by directly evolving visual latents into action latents. VITA removes the need for explicit conditioning modules (e.g., cross-attention), simplifying architectures and improving efficiency. When employing vector representations for both visual and action latents, VITA reduces the flow matching network to a conditioning-free vector-to-vector mapping, enabling a simple MLP-only architecture for complex visuomotor tasks. We show that introducing a latent action space as the flow matching target is critical for successfully learning the vision-to-action flow. To enable effective end-to-end training, we propose flow latent decoding (FLD), which bridges the training–inference gap between encoder-based latents and ODE-generated latents. FLD may serve as a principled approach that can be applied to other generative tasks and latent action policies. Extensive experiments demonstrate that VITA achieves state-of-the-art efficiency and performance on both simulated and real-world tasks, while converging rapidly and stably with high-precision control.

## 7 ETHICS STATEMENT

All experiments were conducted in simulation environments or on standard robotic platforms without involving human subjects, sensitive user data, or any form of personal information. Thus, there are no privacy, security, or human participant concerns. The datasets we use are publicly available benchmark datasets, and no proprietary or restricted data were employed. No conflicts of interest or external sponsorships influence the reported findings.

## 8 REPRODUCIBILITY STATEMENT

We take multiple steps to ensure reproducibility of our results. A detailed description of our model architecture, training objectives, and algorithmic choices is provided in the main text. Hyperparameters, training configurations, and ablations are reported in the Appendix. For theoretical derivations (e.g., flow matching formulation), complete proofs and assumptions are included in the supplementary materials. To facilitate replication, we include open-source code with training scripts, evaluation pipelines, and configuration files. All datasets used are publicly available (Robomimic, ALOHA).

## 9 ACKNOWLEDGEMENTS

This work is supported in part by NSF Grants RINGS-2148253, CNS-2203239, ECCS-2413529, and ARO Grant W911NF-2410046.

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

## A  PROOF OF FLD AND FLC EQUIVALENCE

**A.0.0.1  Preliminaries.**  All norms below are vector norms with induced operator norms. We use the ball $B(\boldsymbol{z}_1, r) = \{\boldsymbol{x} : \|\boldsymbol{x} - \boldsymbol{z}_1\| < r\}$. Assumption 1 in the main text holds throughout.

**Lemma 1** (Local bi-Lipschitzness from Jacobian bounds)**.** *For any $\hat{\boldsymbol{z}}_1 \in B(\boldsymbol{z}_1, r)$,*

$$m \, \|\hat{\boldsymbol{z}}_1 - \boldsymbol{z}_1\| \ \leq \ \|\mathcal{D}_a(\hat{\boldsymbol{z}}_1) - \mathcal{D}_a(\boldsymbol{z}_1)\| \ \leq \ L \, \|\hat{\boldsymbol{z}}_1 - \boldsymbol{z}_1\|.$$

*Proof of Lemma 1.* Let $\gamma(s) = \boldsymbol{z}_1 + s(\hat{\boldsymbol{z}}_1 - \boldsymbol{z}_1)$ for $s \in [0, 1]$. The mean-value integral formula gives

$$\mathcal{D}_a(\hat{\boldsymbol{z}}_1) - \mathcal{D}_a(\boldsymbol{z}_1) = \int_0^1 J_{\mathcal{D}_a}(\gamma(s)) \, (\hat{\boldsymbol{z}}_1 - \boldsymbol{z}_1) \, ds.$$

Taking norms and using, for any matrix $J$ and vector $v \neq 0$, $\sigma_{\min}(J)\|v\| \leq \|Jv\| \leq \sigma_{\max}(J)\|v\|$, together with $m \leq \sigma_{\min}(J_{\mathcal{D}_a}(\gamma(s)))$ and $\sigma_{\max}(J_{\mathcal{D}_a}(\gamma(s))) \leq L$ for all $s$, yields the bounds.

*Proof of Theorem 1.* Add and subtract $\mathcal{D}_a(\boldsymbol{z}_1)$ and apply triangle/reverse-triangle inequalities:

$$\|\mathcal{D}_a(\hat{\boldsymbol{z}}_1) - A\| \leq \|\mathcal{D}_a(\hat{\boldsymbol{z}}_1) - \mathcal{D}_a(\boldsymbol{z}_1)\| + \|\mathcal{D}_a(\boldsymbol{z}_1) - A\|,$$

$$\|\mathcal{D}_a(\hat{\boldsymbol{z}}_1) - A\| \geq \|\mathcal{D}_a(\hat{\boldsymbol{z}}_1) - \mathcal{D}_a(\boldsymbol{z}_1)\| - \|\mathcal{D}_a(\boldsymbol{z}_1) - A\|.$$

Invoke Lemma 1 and set $\varepsilon_{\mathrm{AE}} = \|\mathcal{D}_a(\boldsymbol{z}_1) - A\|$ to obtain the two-sided inequality stated in Theorem 1. The minimizer claims follow immediately: if $\varepsilon_{\mathrm{AE}} = 0$, both losses are minimized at $\hat{\boldsymbol{z}}_1 = \boldsymbol{z}_1$; otherwise any minimizer of FLD must satisfy $\|\hat{\boldsymbol{z}}_1 - \boldsymbol{z}_1\| \leq \varepsilon_{\mathrm{AE}}/m$.

**A.0.0.2  Corollary A.1 (squared-loss version).**  Assume $\varepsilon_{\mathrm{AE}} = 0$. Then

$$m^2 \, \|\hat{\boldsymbol{z}}_1 - \boldsymbol{z}_1\|^2 \ \leq \ \|\mathcal{D}_a(\hat{\boldsymbol{z}}_1) - A\|^2 \ \leq \ L^2 \, \|\hat{\boldsymbol{z}}_1 - \boldsymbol{z}_1\|^2.$$

With $\varepsilon_{\mathrm{AE}} = 0$, Lemma 1 gives $m\|\hat{\boldsymbol{z}}_1 - \boldsymbol{z}_1\| \leq \|\mathcal{D}_a(\hat{\boldsymbol{z}}_1) - \mathcal{D}_a(\boldsymbol{z}_1)\| \leq L\|\hat{\boldsymbol{z}}_1 - \boldsymbol{z}_1\|$. Since both sides are nonnegative, squaring preserves the inequalities. Thus the squared FLD objective is sandwiched between $m^2$ and $L^2$ times the squared FLC objective. Consequently, the map $\hat{\boldsymbol{z}}_1 \mapsto \|\mathcal{D}_a(\hat{\boldsymbol{z}}_1) - A\|_2^2$ is locally $L^2$-smooth and $m^2$-strongly convex along latent directions (intuitively, its curvature is controlled by $J_{\mathcal{D}_a}^\top J_{\mathcal{D}_a}$ whose eigenvalues lie in $[m^2, L^2]$).

**A.0.0.3  Corollary A.2 (gradient scaling for squared losses).**  Let $J := J_{\mathcal{D}_a}(\hat{\boldsymbol{z}}_1)$ and assume $\varepsilon_{\mathrm{AE}} = 0$. For the squared losses,

$$\nabla_{\hat{\boldsymbol{z}}_1} \mathcal{L}_{\mathrm{FLD}}^{(2)} = 2 \, J^\top \big(\mathcal{D}_a(\hat{\boldsymbol{z}}_1) - A\big), \qquad \nabla_{\hat{\boldsymbol{z}}_1} \mathcal{L}_{\mathrm{FLC}}^{(2)} = 2 \, (\hat{\boldsymbol{z}}_1 - \boldsymbol{z}_1).$$

Then

$$m^2 \, \big\|\nabla \mathcal{L}_{\mathrm{FLC}}^{(2)}\big\| \ \leq \ \big\|\nabla \mathcal{L}_{\mathrm{FLD}}^{(2)}\big\| \ \leq \ L^2 \, \big\|\nabla \mathcal{L}_{\mathrm{FLC}}^{(2)}\big\|.$$

Use $\|J^\top y\| \in [m\|y\|, L\|y\|]$ (by singular-value bounds) with $y = \mathcal{D}_a(\hat{\boldsymbol{z}}_1) - A = \mathcal{D}_a(\hat{\boldsymbol{z}}_1) - \mathcal{D}_a(\boldsymbol{z}_1)$, and Lemma 1 to get $m\|y\| \leq \|J^\top y\| \leq L\|y\|$ and $m\|\hat{\boldsymbol{z}}_1 - \boldsymbol{z}_1\| \leq \|y\| \leq L\|\hat{\boldsymbol{z}}_1 - \boldsymbol{z}_1\|$. Multiplying the bounds yields $m^2\|\hat{\boldsymbol{z}}_1 - \boldsymbol{z}_1\| \leq \|J^\top y\| \leq L^2\|\hat{\boldsymbol{z}}_1 - \boldsymbol{z}_1\|$. Since $\|\nabla \mathcal{L}_{\mathrm{FLC}}^{(2)}\| = 2\|\hat{\boldsymbol{z}}_1 - \boldsymbol{z}_1\|$, this gives the stated inequality (up to the common factor 2). It follows that step-size sensitivity is governed by the squared condition number $(L/m)^2$.

## B  DISCUSSIONS

### B.1  DIMENSIONALITY MATCHING FOR VISION-TO-ACTION FLOW

A constraint of flow matching is that the source and target must have identical dimensionality. In visuomotor contexts, the visual latent representations ($\boldsymbol{z}_0$) are typically much higher-dimensional than raw action chunks ($A$). A naive solution would be to down-sample the visual representation to match the action dimensionality, which, nevertheless, leads to significant information loss and poor performance, particularly when the dimensional gap is large.

Therefore, we adopt the opposite strategy: we map the raw action chunks into a higher-dimensional latent space that matches the dimensionality of the visual latent representations.

A naive approach is to use fixed linear transformations. When the action dimension is smaller than the latent dimension, we construct a lossless mapping by embedding the actions into the higher-dimensional latent space through zero-padding. The inverse mapping simply discards the padding. Our experiments show that action representations produced by such transformations are insufficient for learning reasonable flow matching policies. We found that learning well-structured action latent spaces via autoencoders as the target distributions for flow matching is crucial for the success of vision-to-action flow. We develop an action encoder ($\mathcal{E}_a$), which does not simply remap dimensions, but also learns structured latent action spaces, making the complex flow from vision to action more tractable.

Table 4: Task SR (%) on `ThreadNeedle` with different action up-sampling strategies.

| Up-sampling Strategy | SR (%) |
| --- | --- |
| Zero-Padding | 0 |
| Action AE (w/o FLD) | 0 |
| Action AE (w/ FLD) | **92** |

## B.2 ABLATION OF FROZEN TARGET FOR FLOW MATCHING

Learning a flow when the target distribution lies in a learned latent space is inherently challenging. Jointly optimizing the flow and the latent encoder creates a moving target, as the latent space continually shifts during training; it also introduces a training-inference gap: flow matching does not guarantee that ODE-generated latents are decodable during inference, since the decoder is trained to reconstruct encoder-based latents.

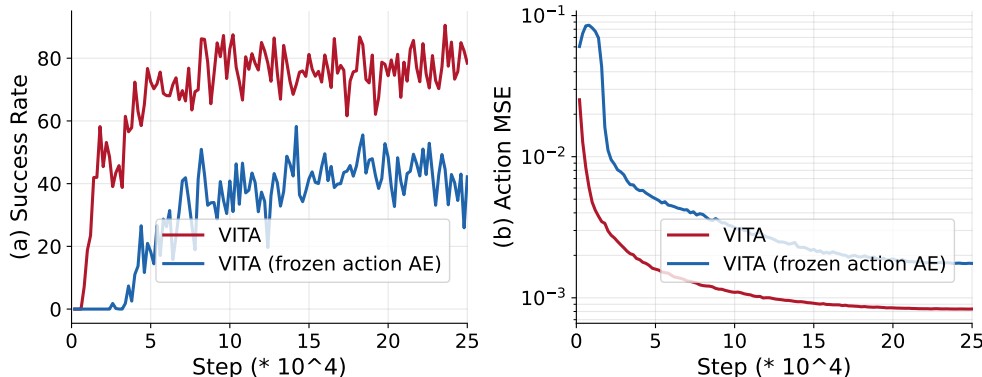

Figure 8: Success rates (left) and log-scaled action MSEs (right) comparing end-to-end VITA training with VITA using a frozen action AE on `ThreadNeedle`. EMA = 0.9.

A natural solution is to follow the strategy in latent diffusion models (Rombach et al., 2022): pre-train the latent space on large-scale data and freeze it when training the flow. We evaluate this approach by pre-training the action autoencoder for 100k steps using a reconstruction loss, $\mathcal{L}_{AE}$, then freezing it while training the VITA policy for 25k steps (same as VITA trained from scratch). As shown in Figure 8, this frozen-latent setup yields poor online success rates and high offline MSEs that plateau early. In contrast, end-to-end VITA training enabled by FLD performs substantially better. The underlying issue is that, unlike image generation, robotics action data is sparse and limited; pure latent pre-training produces a weak representation that cannot be improved once the latent space is frozen.

## B.3 CONTRASTIVE LATENT ALIGNMENT

Section 4.4 introduced two key objectives, FLD and its surrogate FLC, which are both designed to prevent latent space collapse and enable effective VITA learning. Empirically, we find that FLD

alone outperforms FLC alone, as it provides a more direct training signal by anchoring the generation process to the ground-truth actions. However, the most robust performance is achieved by combining them, which enforces consistency in both latent action space and raw action space.

Additionally, motivated by the ability of contrastive learning to improve representations and prevent latent collapse (Liu et al., 2024a), we introduced a contrastive loss between vision and action latents (Radford et al., 2021). We show that this objective can further boost performance beyond FLD and FLC on some tasks by encouraging the model to learn representations where the similarity between corresponding vision-action pairs is maximized, while the similarity between non-corresponding pairs is minimized.

We employ InfoNCE (Noise-Contrastive Estimation) for contrastive learning between vision and action. For a given batch of size $N$, the vision latent $z_{0,i}$ and action latent $z_{1,i}$ from the same data sample are treated as a positive pair. All other non-corresponding combinations $(z_{0,i}, z_{1,j})$ where $i \neq j$ are considered negative pairs. The loss aims to maximize the similarity of positive pairs while minimizing the similarity of negative pairs. The symmetric InfoNCE loss is defined as:

$$\mathcal{L}_{\text{contrastive}} = -\frac{1}{2N} \sum_{i=1}^{N} \left[ \log \frac{\exp(\text{sim}(z_{0,i}, z_{1,i})/\tau)}{\sum_{j=1}^{N} \exp(\text{sim}(z_{0,i}, z_{1,j})/\tau)} + \log \frac{\exp(\text{sim}(z_{1,i}, z_{0,i})/\tau)}{\sum_{j=1}^{N} \exp(\text{sim}(z_{1,i}, z_{0,j})/\tau)} \right]$$

where $\text{sim}(\cdot, \cdot)$ denotes the cosine similarity between L2-normalized feature vectors and $\tau$ is a temperature hyperparameter.

As shown in Figure 9, the contrastive objective, when used alone, is insufficient for effective VITA learning and underperforms VITA with only the FLD loss (see Figure 6). However, it provides additional performance gains when combined with the FLD and FLC losses on some tasks, as it helps create a more robust and well-structured latent space.

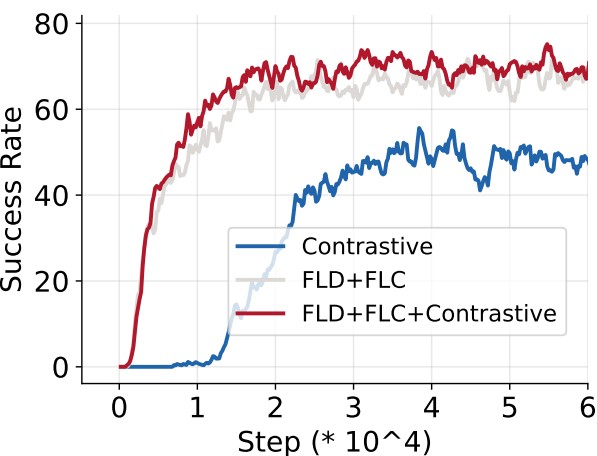

Figure 9: Success rates using different VITA learning objectives with or without contrastive losses.

### B.4 ABLATION OF THE VARIATIONAL ACTION AUTOENCODER

In our experiments, a deterministic autoencoder (AE) learns the target latent space for actions. To investigate the effect of imposing a prior on this latent space, we conducted an ablation study replacing the AE with a variational autoencoder (VAE) (Kingma et al., 2013). This change introduces a KL divergence regularization term, weighted by $\lambda_{\text{KL}}$, which encourages the encoder's posterior output, $q(z_1 \mid A)$, to match a standard normal prior:

$$\mathcal{L}_{\text{KL}} = D_{\text{KL}}\big(q(z_1 \mid A) \,\big\|\, p(z_1)\big), \qquad p(z_1) = \mathcal{N}(0, I).$$

As shown by the training curves in Figure 10(a) and Figure 21, incorporating this variational objective with various weights ($\lambda_{\text{KL}}$) can degrade model performance.

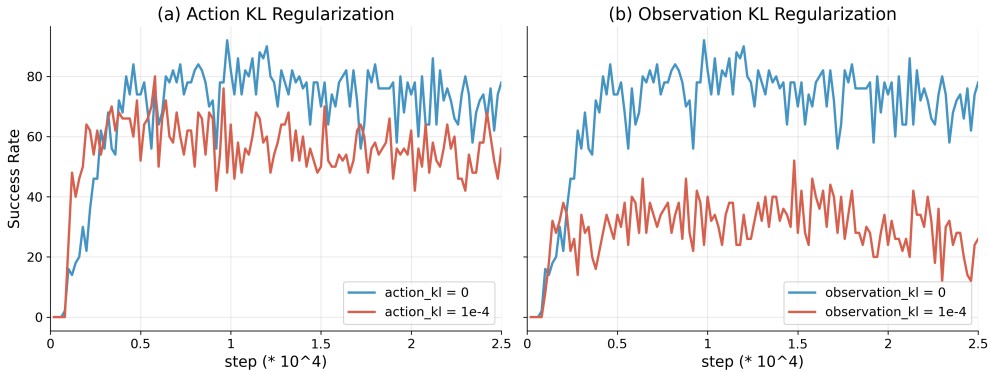

Figure 10: Ablation of VAE for action (a) and vision (b) on `ThreadNeedle`.

### B.5    ABLATION OF THE VARIATIONAL VISION ENCODER

As depicted in our main architecture (Figure 2), the source latent variable $z_0 = \mathcal{E}_v(O) \in \mathbb{R}^{D_{\text{latent}}}$ is produced by a deterministic vision encoder. We performed a similar ablation to assess the impact of introducing stochasticity to visual encoding. Specifically, we replaced the deterministic encoder with a variational one that models the posterior $q(z_0|O)$ and introduces a corresponding KL divergence loss (weighted by $\lambda_{\text{KL}}^{\text{obs}}$):

$$\mathcal{L}_{\text{KL}}^{\text{obs}} \;=\; D_{\text{KL}}\big(q(z_0 \mid O) \,\big\|\, p(z_0)\big), \qquad p(z_0) = \mathcal{N}(0, I).$$

As shown in Figure 10(b), employing a VAE for the vision encoder drastically degrades performance.

Using a VAE to model the source distribution introduces stochasticity, which is often desirable for generative tasks that emphasize diversity (Liu et al., 2024a). Similarly, stochasticity in DP and FM helps policies model action multi-modality (Chi et al., 2023). However, we found that making the visual latent encoding stochastic degrades VITA performance. In robotics tasks that demand extremely high precision, such as `ThreadNeedle`, where millimeter-level errors can cause complete failure, variational objectives tend to blur latent representations, discarding critical visual details. As a result, deterministic vision encodings yield substantially better precision and performance. We discuss the precision requirements of robotics tasks in greater detail in Appendix B.8.

### B.6    ABLATION OF NETWORK ARCHITECTURE

VITA, when using vector representations for both vision and actions, reduces the flow matching network to a conditioning-free vector-to-vector mapping. We find that even an MLP-only architecture can successfully learn challenging, high-precision visuomotor tasks, including bimanual manipulation with active vision on AV-ALOHA. To the best of our knowledge, VITA is the first visuomotor policy to master such complex tasks using MLPs.

#### B.6.1    FM USING MLP

We evaluate FM using the same 4-layer MLP architecture as VITA (with a similar parameter count of ~30M). Concretely, we remove self-attention from the DiT-based FM network and retain only AdaLN (which is also MLP-based) for visual conditioning. However, the MLP-only FM fails to learn effective policies: on `PushT`, the reward remains around 0.4 and success rate remains 0% even after 100K training steps, as the task requires high-precision control (successful only when reward exceeds 0.95). In contrast, VITA and transformer-based FM achieve 88% and 83% success, respectively. This failure arises because an MLP is insufficient for processing noisy action chunks and integrating visual conditioning at each denoising step, leading to poor control precision, as shown in Figure 11(b), where action MSEs of MLP-based FM plateau while VITA yields significantly lower MSEs. Since success rates are sparse on `PushT`, we also report online reward curves (see Figure 11(a)) to more clearly compare MLP-based FM and VITA.

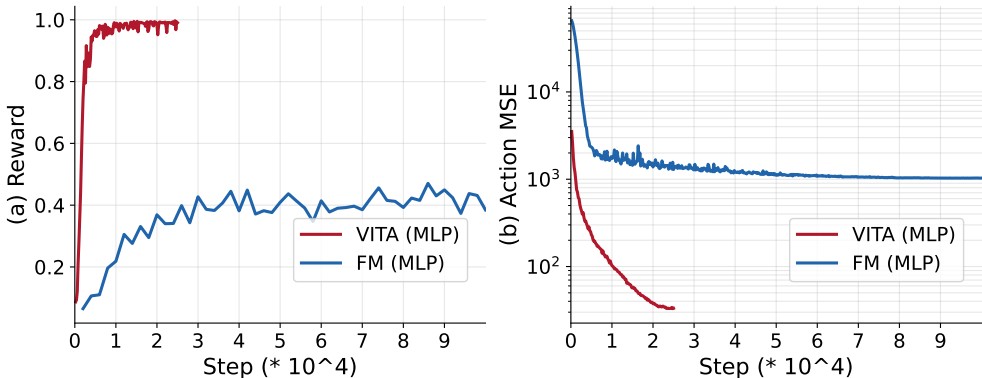

Figure 11: Reward curves and the action MSEs (log-scaled) of MLP-only VITA and MLP-only FM on `PushT`. FM learning is ineffective because lack of precision, and performs poorly online, necessitating effective architectures to process action chunks and fuse in conditions.

The fundamental reason is that **conventional FM learns a transport from an unstructured isotropic Gaussian prior to a highly structured action distribution**, which places a heavy burden on the flow network to simultaneously denoise, integrate visual conditioning, and recover precise control signals at each step. This requires strong architectural inductive biases (e.g., transformers or U-Nets) to effectively model high-dimensional noisy action chunks. In contrast, VITA operates between two structured and semantically aligned latent spaces, where visual latents already encode coarse action-relevant structure. As a result, the flow only needs to perform incremental refinement rather than reconstruct actions from noise, allowing a lightweight MLP architecture to suffice without sacrificing control precision.

### B.6.2 VITA USING TRANSFORMERS

Additionally, we show that VITA is not limited to vector-based features or MLP. We evaluate VITA using grid-based features (in particular, $9 \times 512$ spatial features obtained via ResNet-18), and use transformer for the flow matching network. The flow matching network does not require any conditioning for spatial tokens such as costly cross-attention compared to FM using transformers.

We evaluate VITA on multiple challenging tasks, demonstrating that VITA yield high success rates (see Figure 12) while retaining the efficiency gains (see Table 1 and Appendix B.7).

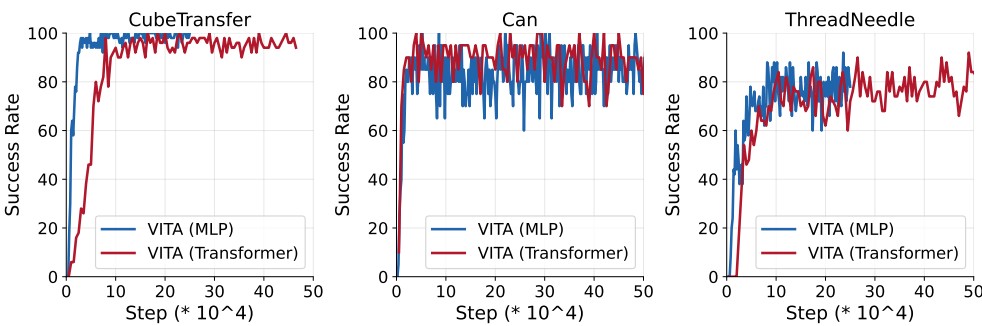

Figure 12: Comparing success rates of VITA using MLP and vector-based features and VITA using transformers and grid-based features.

### B.7 Extended Efficiency Metrics

#### B.7.1 Training Time & Space Overheads of Flow Latent Decoding

Recall that we introduce FLD during training to enable effective end-to-end learning, which requires solving 6-step ODEs. Although this inevitably adds some training-time overhead, we trade a modest increase in training cost for significantly more efficient inference, which is critical for real-time robotic deployment. For MLP-based VITA, the training time increases from 0.677 ms/chunk (without FLD) to 0.740 ms/chunk (with FLD), corresponding to only a 9.3% overhead. For transformer-based VITA, the cost rises from 0.766 ms/chunk to 0.951 ms/chunk, a 24.1% overhead. A similar trend appears for GPU memory usage: for MLP-based VITA, peak training memory increases from 2716.62 MiB (without FLD) to 2835.86 MiB (with FLD), a relatively small 4.4% increase. Despite these modest training-time and memory overheads, FLD enables stable end-to-end optimization and yields markedly improved inference-time efficiency, a critical requirement for real-time robotic deployment. Additionally, VITA maintains the lowest training-time memory usage compared to all baselines even with FLD enabled, while achieving comparable training latency.

#### B.7.2 Training Memory Usage

We have reported the absolute peak memory usage of each policy architecture in Table 1, measured via `torch.cuda.max_memory_allocated` for fair comparison. Importantly, we exclude the shared visual encoder, which transforms raw RGB images into visual latents and is used identically across all methods. This observer accounts for approximately 1732.5 MiB of peak GPU memory across all methods, and does not help differentiating memory usage across policies. For instance, after removing observer cost, VITA (MLP) peaks at 333.86 MiB, FM (MLP, AdaLN) peaks at 413.95 MiB, and FM (Transformer, Cross-Attn) reaches 529.16 MiB. These numbers reflect the true architectural and conditioning differences in the policy modules themselves. Consequently, VITA (MLP) reduces peak memory usage by 19.4% relative to FM (MLP, AdaLN), 18.6% relative to FM (Transformer, AdaLN), and 36.9% relative to FM (Transformer, Cross-Attn), highlighting its inference-time efficiency gains from a conditioning-free formulation.

In Table 5, we report the absolute peak memory usage during training for each policy. As with inference, we ablate the memory consumed by the shared visual encoder, and record peak memory after vision encoding. For VITA, we also include the peak usage observed during the `flow_latent_decoding` phase, which accounts for the highest memory load. This choice ensures that the reported number reflects the true upper bound of VITA training footprint with FLD.

Table 5: Comparison of the conditioning parameter overhead, training-time cost, and training-time memory usage of VITA and baselines, grouped by the type of visual latents used ("Vector" or "Grid" based). Metrics include (i) parameters introduced solely by conditioning modules, (ii) training time per chunk (ms), and (iii) peak GPU memory during training (MiB).

| Visual | Model | Architecture | Conditioning | Cond. Params (M) | Time | Memory |
|--------|-------|--------------|--------------|------------------|------|--------|
| **Vector** | **VITA** | **MLP** | *N/A* | **0.00** | 0.740 | **2835.86** |
| | FM | MLP | AdaLN | 11.82 | **0.664** | 2926.60 |
| | FM | Transformer | AdaLN | 6.58 | 0.697 | 3071.88 |
| | FM | U-Net | FiLM | 11.33 | 0.782 | 3676.38 |
| | DDPM | U-Net | FiLM | 9.49 | 0.779 | 3643.04 |
| **Grid** | VITA | Transformer | *N/A* | **0.00** | 0.951 | **2977.10** |
| | FM | Transformer | Cross-Attn | 4.47 | **0.812** | 3585.06 |

### B.8 Control Precision vs. Sampling Stochasticity

#### B.8.1 VITA with Sampling Stochasticity

Since stochastic visual encodings degraded performance (as discussed in Section B.5), and we hypothesized this was due to precision loss from blurred latent representations, we further investigated the trade-off between stochasticity and control performance. We examined VITA variants with different sources of sampling stochasticity. Introducing dropout in the network, or variance ($\sigma$) in

flow matching, where $\sigma$ injects Gaussian noise along the interpolation path, consistently reduced performance. In contrast, adding covariance to the source distribution produced results comparable to deterministic encoding. Likewise, using a variational objective in the action autoencoder performed similarly to the deterministic action autoencoder, whereas applying a variational objective to the image encoder significantly harmed performance (see Figure 10).

These findings resonate with our observation that DP underperforms FM or VITA on ALOHA tasks that require high precision. DP is based on an SDE (stochastic differential equation) while FM uses a deterministic ODE formulation. FM introduces stochasticity only through sampling the Gaussian prior, VITA goes even further by removing the Gaussian prior sampling, instead using a visually grounded and deterministic initial state for the flow. Together, these results can suggest a broader trend: for fast and precise real-time control, reducing stochasticity can be beneficial in, e.g., speeding up convergence, producing more precise and faster policies.

### B.8.2 Precision-Critical Robot Control Tasks

We followed DP and ACT implementation from LeRobot (Cadene et al., 2024). However, DP and ACT perform poorly ( 40% SRs compared to 80%-90% of VITA and FM) on ALOHA tasks such as `ThreadNeedle` and `PourTestTube`, which demand high precision.

This section examines the stringent success criteria of these tasks and explains why even small action errors lead to failures. We further show that VITA and FM achieve substantially better action precision with far fewer training steps, which contributes to their superior performance on these precision-demanding tasks.

Stochasticity helps model action multimodality, which has been considered a key factor in the success of generative policies (Chi et al., 2023). However, VITA does not involve stochastic sampling by default. Recently, Pan et al. (2025) show that multimodality alone does not explain the success of generative policies. We further show that precision can be more critical for many robotics tasks, and deterministic sampling can help improve precision.

**Success Criteria.** We observed that DP and ACT learn reasonable trajectories, but small millimeter-level errors lead to binary failures. For example, because success requires completing all five subtasks on `ThreadNeedle`, failing at the third stage still counts as a full failure, yielding low SRs when the average reward exceeds 3. In contrast, as depicted in Figure 13, both VITA and FM learn sufficiently precise control to complete the final subtask with fewer training steps, achieving high success rates on these tasks.

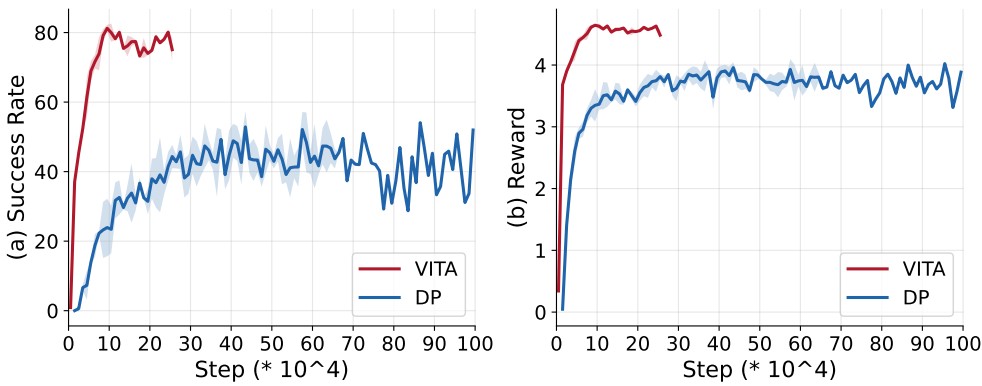

Figure 13: Success rate and reward curves of VITA and DP on `ThreadNeedle`.

As shown in Figure 14, DP can complete multiple subtasks yet fail the episode due to millimeter-level precision errors, such as threading a needle through a very small opening. In contrast, we find that VITA and FM exhibit higher control precision on these tasks.

**Action Precision.** We now examine why VITA and FM overall achieve higher action precision. As shown in Table 3, VITA outperforms all baselines on most tasks, and FM achieves SRs comparable to VITA. To better understand this, we analyze the offline action MSE during training for VITA, FM, DP,

Failure

Success

Figure 14: A failure and a success case on `ThreadNeedle`. DP may complete most subtasks but still fail the final insertion due to millimeter-level errors.

and ACT in Figure 15. We observe that VITA and FM consistently converge to lower MSEs, whereas ACT plateaus at substantially higher errors, and DP uses much more training steps to converge. This trend aligns with prior findings (e.g., on image generation) showing that flow matching methods enjoy faster convergence (Lipman et al., 2024) and can achieve higher generation fidelity (Gupta & Taiwade, 2025).

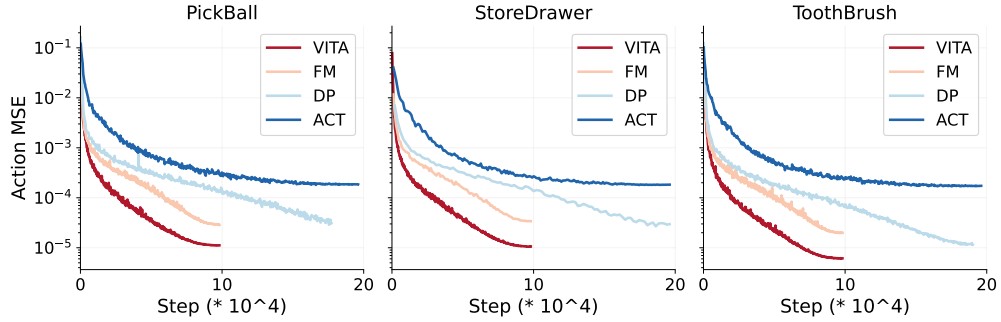

Figure 15: Comparison of action MSE on three real-world single-arm ALOHA tasks.

### B.9 AUTOENCODER LOSS SELECTION

We utilize the L1 loss for the autoencoder loss, $\mathcal{L}_{AE}$. We found it outperforms the L2 loss, which is prone to mode-averaging and can result in blurry action reconstruction.

### B.10 SUCCESS RATE CURVES DURING VITA LEARNING

VITA exhibits efficient and stable learning on AV-ALOHA and `PushT` (Figure 16) and Robomimic tasks (Figure 17).

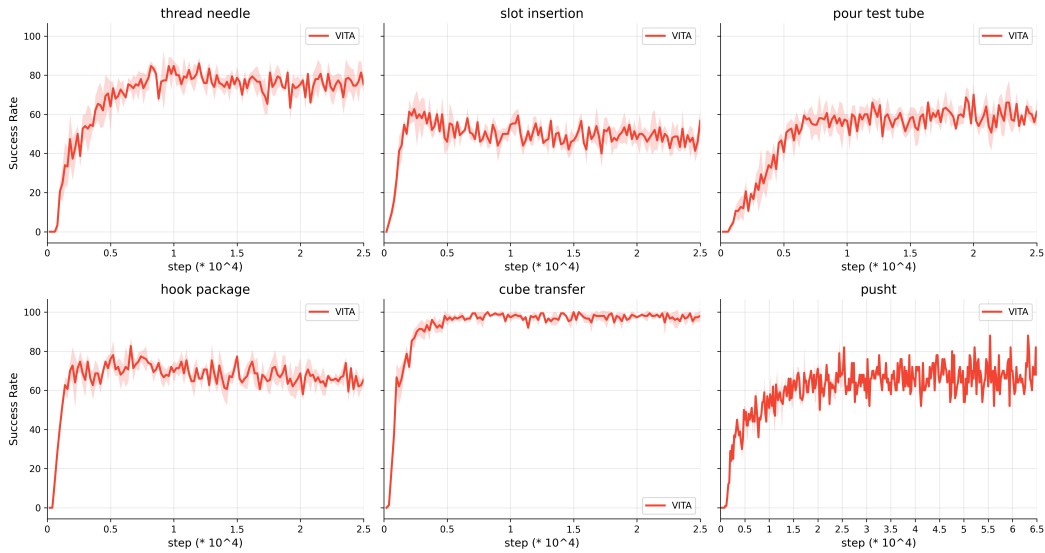

Figure 16: Success rate curves of VITA training on six tasks (five AV-ALOHA + `PushT`). The curves are mean across three random seeds; the shaded region is $\pm 1$ standard deviation.

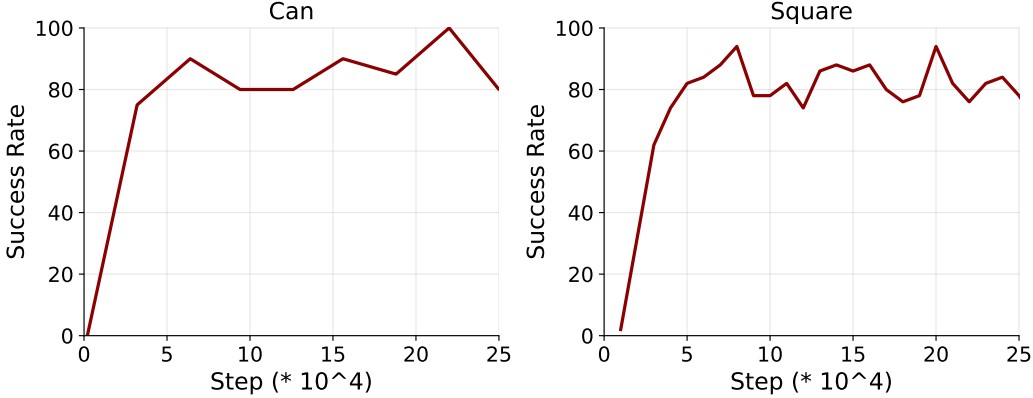

Figure 17: Success rate curves of VITA training on two Robomimic tasks.

### B.11 LATENT COLLAPSE

Our motivation for FLD stems from two key observations: first, the multi-stage pipeline inspired by latent diffusion utilizing a pre-trained autoencoder is suboptimal for action data (Section B.2); and second, joint training of the flow and target latent space leads to latent collapse (Section 5.3).

Notably, very recent work Wang & Zhang (2025) resonates with these findings, and independently identified a similar failure mode in latent diffusion for image generation. They observed that the diffusion objective can undermine representation quality during end-to-end training, and proposed Diffusion as Self-Distillation (DSD) Wang & Zhang (2025) to stabilize latent representation learning. While FLD and DSD employ different mechanisms in distinct domains, both methods corroborate the necessity of end-to-end optimization while stabilizing latent representation learning.

## C EXTENDED RELATED WORKS

**Imitation Learning for Visuomotor Policy.** Imitation learning enables robots to learn complex behaviors by mimicking expert demonstrations. Behavioral cloning, a prominent imitation learning

paradigm, frames this as a supervised learning problem, learning a policy that maps observations to actions (Zhao et al., 2023; Lee et al., 2024). Recent advancements in behavioral cloning have widely adopted generative modeling, which learns a conditional distribution of actions given an observation. This category includes policies based on conditional variational autoencoders (CVAEs) (Zhao et al., 2023; Lee et al., 2024), as well as diffusion (Dasari et al., 2024; Chi et al., 2023) and flow matching (Zhang & Gienger, 2024; Zhang et al., 2025). Autoregressive methods, which tokenize actions and frame policy learning as a sequence modeling task, are also well-studied. These methods predict action tokens sequentially, using next-token prediction (Fu et al., 2024a), next-scale prediction (Gong et al., 2024), or bi-directional prediction (Su et al., 2025). Generative models ubiquitously require extra conditioning modules (e.g., cross-attention (Dasari et al., 2024), AdaLN (Dasari et al., 2024), FiLM (Perez et al., 2018; Chi et al., 2023)) to inject observations at each step of the generation process. Furthermore, generative and autoregressive methods commonly employ large, expressive networks such as U-Nets or transformers to succeed on complex, high-dimensional robotics tasks. VITA reduces this complexity by formulating the policy as a noise-free and conditioning-free vision-to-action flow.

**Diffusion and Flow matching for Generative Modeling**. Diffusion (Ho et al., 2020), grounded in stochastic differential equations (SDEs), generates complex data distributions by sampling from a simple source distribution (typically Gaussian) and iteratively denoising it to the target distribution (Sohl-Dickstein et al., 2015). Flow matching (Lipman et al., 2023; Liu et al., 2022b) has been proposed to enable faster training and sampling (Liu et al., 2022b; Tong et al., 2024; Esser et al.; Lipman et al., 2023) by modeling the map between source and target distributions with an ordinary differential equation (ODE). Both diffusion and flow matching models have shown strong performance across diverse generative tasks, such as image generation (Rombach et al., 2022; Peebles & Xie, 2023; Ma et al., 2024; Zhang et al., 2023; Liu et al., 2024b; Ren et al., 2024b), video generation (Ho et al., 2022; Li et al., 2023), and visuomotor policies (Chi et al., 2023; Dasari et al., 2024; Black et al.; Liu et al., 2024c). Unlike diffusion, flow matching theoretically places no constraints on the choice of source distribution (Tong et al., 2024), and a few works have explored leveraging this property to learn the direct transport within the same modality (Albergo & Vanden-Eijnden, 2022; Tong et al., 2023b), e.g., for image-to-image generation tasks (Fischer et al., 2023; Liu et al., 2022a). Recently, Liu et al. (2024a) and He et al. (2025) extended this to more challenging cross-modal generation between text and image. VITA focuses on learning to bridge vision and action for visuomotor control, where the target modality, action, has sparser data and lacks semantic structures, compared to text or images, presenting unique challenges. Different from flow matching for image generation, which typically pre-trains and freezes the image autoencoder when learning flow matching or diffusion models for image generation (Rombach et al., 2022; He et al., 2025; Liu et al., 2024a), VITA resorts to a fully end-to-end pipeline training to effectively learn the latent action space from limited and sparse action data. Furthermore, to enable effective joint training of flow matching and target latent spaces, we propose flow latent decoding to backpropagate action reconstruction losses through the latent action generation process (ODE solving steps) during training. To the best of our knowledge, **VITA is the first policy to jointly learn flow matching and a latent action space end-to-end.**

## D   SIMULATED AND REAL-WORLD TASKS

To comprehensively evaluate the effectiveness of VITA across varying levels of difficulty and action dimensionality, we conduct extensive experiments on both single-arm and bimanual manipulation tasks. The action dimensionality spans from 2 to 21, and the tasks include both short- and long-horizon settings. Overall, AV-ALOHA tasks are particularly challenging due to their 21D action spaces, non-stationary observations introduced by the active-vision camera, and long-horizon, precision-demanding task structure (see Figure 18 for real-world examples). Single-arm ALOHA tasks are also challenging due to randomness, such as varying object types and object poses.

The specifications for each dataset are shown in Table 6. Following the practice of AV-ALOHA (Chuang et al., 2024), we train all policies at 8.33 FPS (25/3) for simulated AV-ALOHA tasks and at 11 FPS for real AV-ALOHA tasks, and interpolate to 25 FPS and 33 FPS, respectively, for inference.

| Dataset | State Dim | Action Dim | FPS | Image Size | Camera |
|---------|-----------|------------|-----|------------|--------|
| AV-ALOHA (Sim) | 21 | 21 | 25 | 240×320 | `zed_cam_left` |
| AV-ALOHA (Real) | 21 | 21 | 33 | 240×320 | `left_eye_cam` |
| ALOHA (Real) | 7 | 7 | 33 | 240×320 | `overhead_cam,`
`right_wrist_cam` |
| Robomimic | 43 | 7 | 20 | 256×256 | `agentview_image` |
| PushT | 2 | 2 | 20 | 96×96 | `image` |
| CloseBox | 9 | 9 | 20 | 256×256 | `head_cam` |

Table 6: Comparison of dataset specifications.

## D.1 AV-ALOHA SIMULATION TASKS.

`CubeTransfer`: Pick up a red cube with the right arm and transfer it to the left arm (200 episodes).

`SlotInsertion`: Use both arms to pick up a green stick and insert it into a pink slot (100 episodes).

`HookPackage`: Use both arms to pick up a red box and hook it onto a blue wall-mounted hook (100 episodes).

`PourTestTube`: Pick up two test tubes and pour a small red ball from one into the other (100 episodes).

`ThreadNeedle`: Pick up a green needle and thread it through the hole of a pink object (200 episodes).

## D.2 ALOHA REAL TASKS

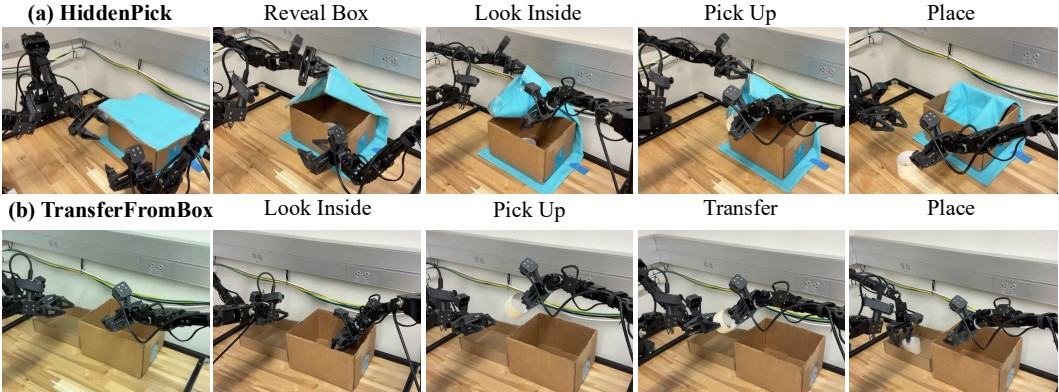

Figure 18: Autonomous rollouts of VITA on two challenging real-world AV-ALOHA tasks, `HiddenPick`, and `TransferFromBox`.

### D.2.1 BIMANUAL MANIPULATION WITH ACTIVE VISION

To evaluate the effectiveness of VITA in real-world settings, we deploy the policy on two challenging bimanual manipulation tasks using AV-ALOHA (Chuang et al., 2024). Both tasks are long-horizon and contain multiple stages to succeed. Examples of autonomous rollouts are shown in Figure 18. Both tasks require precise and coordinated control of three arms, including one arm carrying an active vision camera, and two arms for bimanual manipulation.

`HiddenPick`: Lift and open a fabric cover from a box, then pick an object from inside (50 episodes).

`TransferFromBox`: Pick an object from a box with the right arm, transfer it to the left arm, and place it in another box (50 episodes).

Table 7: SRs of VITA on two real-world bimanual manipulation tasks with active vision on AV-ALOHA. Each task is decomposed into three subtasks, and SRs are reported per subtask.

| | HiddenPick | | | TransferFromBox | | |
|---|---|---|---|---|---|---|
| | Reveal | Pick | Place | Pick | Transfer | Place |
| **VITA** | 1.00 | 0.65 | 0.65 | 1.00 | 0.95 | 0.90 |

### D.2.2 SINGLE-ARM MANIPULATION TASKS

Each single-arm ALOHA task consists of multiple stages, and includes substantial environment randomization as detailed below.

`PickBall`: Pick up a ball, then place it into the box (50 episodes). Both the ball and the target box appear in varying positions.

`ToothBrush`: Pick up the toothbrush from the side slot of the toothbrush cup, lift it up, and place it into the toothbrush cup (50 episodes). The cup location and the orientation of the side slot are randomized.

`StoreDrawer`: Pick up the object, put it in the drawer, and close the drawer (100 episodes). We randomize the shapes and colors of the objects as well as the positions and partial openings of the drawers. We also evaluate the out-of-distribution success rates for unseen combinations of object colors and shapes (see Appendix E.2).

### D.3 ROBOMIMIC TASKS

Robomimic is a benchmark of single-arm imitation learning tasks Mandlekar et al. (2021). We adapt the environment for compatibility with the LeRobot (Cadene et al., 2024) codebase. The robot state includes arm joint positions (encoded with $\sin$ and $\cos$), joint velocities, end-effector pose, gripper finger positions, and gripper finger velocities. The action space consists of six values for delta position control of the end-effector pose and one value for the absolute position of the gripper.

`Square`: Pick up a square nut and insert it onto a matching square peg (175 episodes).

`Can`: Pick up a red can and place it into a box (192 episodes).

### D.4 OTHER TASKS

`PushT`: Push a 2D T-shaped object into a matching T-shaped target region on the plane; achieving a coverage score $> 0.95$ is deemed a success (200 episodes).

`CloseBox`: Close the lid of a paper box by manipulating the flap (200 episodes).

## E ADDITIONAL EXPERIMENT RESULTS

### E.1 ROBUSTNESS TO ONLINE PERTURBATIONS

As depicted in Figure 19, VITA demonstrates strong robustness to online perturbations during real-time control.

### E.2 GENERALIZATION TO UNSEEN OBJECTS

As shown in Figure 20, we train the `StoreDrawer` task using 7 objects. Table 3 shows that VITA achieves the highest success rate on these in-distribution objects. To further evaluate generalization, we introduce four unseen test objects with novel geometries, such as a triangular prism and a star-shaped block. On this out-of-distribution (OOD) set, VITA succeeds in picking all objects and storing them in the drawer.

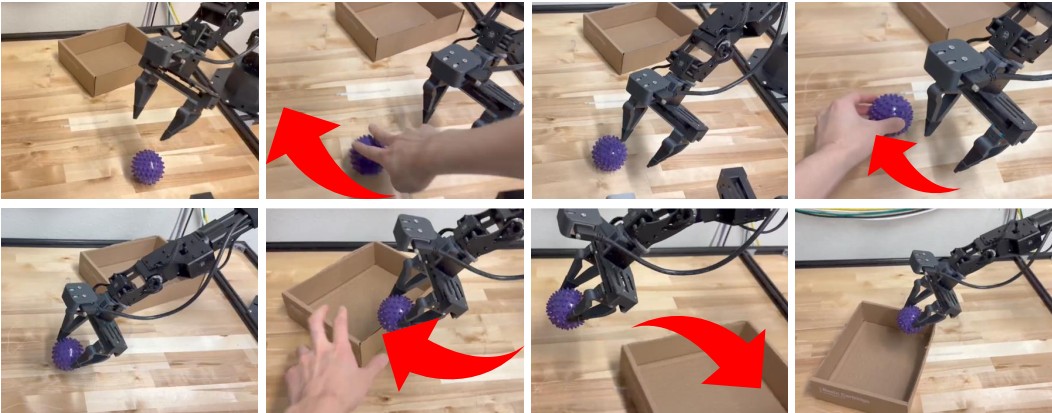

Figure 19: Robustness to online perturbations during inference on `PickBall`. We manually move the ball multiple times before the pick, and move the box multiple times after the pick. In both cases, the arm continues to adjust in real time and successfully reaches the correct ball and box positions.

Table 8: OOD SRs on four unseen objects for the `StoreDrawer` task.

| VITA | DP | FM | ACT |
|------|------|------|------|
| 4/4 | 4/4 | 3/4 | 2/4 |

### E.3 ONE-STEP DENOISING

One-step generation has been widely explored in diffusion and flow matching models (Song et al., 2023; Zhang et al., 2025). While we employ OT-CFM (Tong et al., 2023a) with a 6-step Euler ODE solver in this work, VITA is compatible with other flow matchers, including one-step methods such as MeanFlow (Geng et al., 2025; Sheng et al., 2025). Integrating these methods enables further inference acceleration, albeit at the expense of generation quality. For instance, applying MeanFlow to the `PushT` task yields approximately 2x faster inference, but reduces the success rate from 88% to 74%. Exploring these extreme efficiency improvements while mitigating action precision degradation remains a promising direction for future work.

## F   TRAINING

In each plot of Figure 21, we tune a single hyperparameter while holding all other hyperparameters at the default (see Table 9), and visualize the success rates over training steps. With these experimental results, a robust configuration on `ThreadNeedle` includes moderate weights for both AE and FLD (typically in $[0.5, 1.0]$), moderate FLC, minimal contrastive penalties, and no KL regularization on action and visual latents.

## G   IMPLEMENTATIONS

### G.1   VITA

VITA encodes observations into a latent vector $z_0$ using a ResNet-18 backbone. The flow network, which learns the mapping from $z_0$ to $\hat{z}_1$, is implemented as an MLP. We use a 6-step Euler ODE solver. The latent action $\hat{z}_1$ is translated to action chunks by a lightweight MLP-based action decoder. A summary of all hyperparameters and loss weights is provided in Table 9.

**(a) Objects used during Training**    **(b) Objects used during OOD Evaluation**

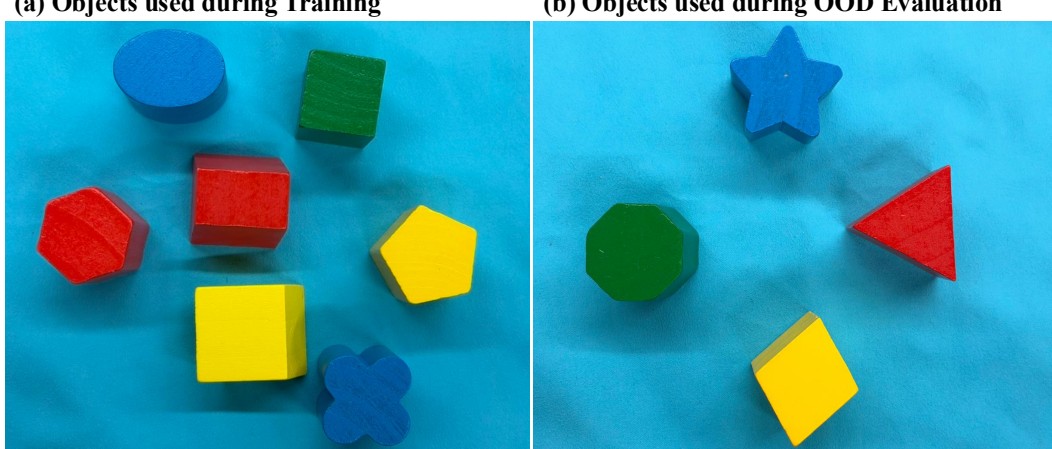

Figure 20: OOD evaluation on `StoreDrawer`. Training uses 8 in-distribution object–color combinations (left). Evaluation uses four unseen objects with novel shapes (right), including a triangular prism and a star-shaped block.

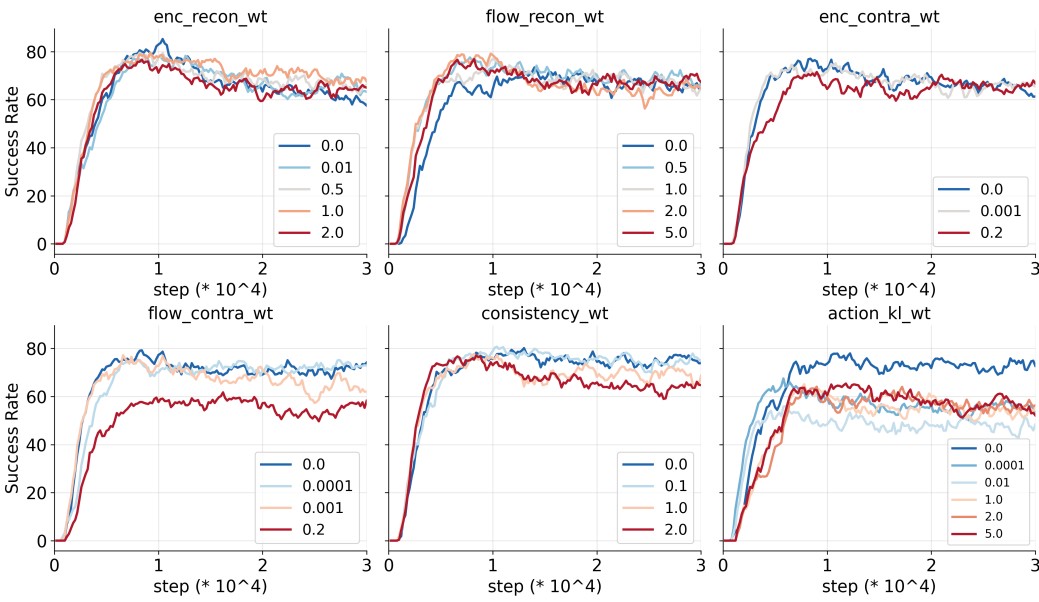

Figure 21: Hyperparameter ablations on `ThreadNeedle`. Each sub-plot varies a single coefficient while holding all others at their default values in Table 9. Top: action encoder reconstruction weight, flow latent decoding weight, action encoder contrastive weight. Bottom: latent flow contrastive weight, latent flow consistency weight, action KL weight.

### G.2    FLOW MATCHING POLICY

The FM policy learns a velocity field using a transformer backbone, with AdaLN for conditioning. A 6-step Euler solver is used for solving ODEs. The hyperparameters and loss weights are summarized in Table 10.

### G.3    DIFFUSION POLICY

We follow the DP (Chi et al., 2023) implementation in LeRobot (Cadene et al., 2024). The core of the policy is a U-Net noise predictor, which uses FiLM (Perez et al., 2018) for conditioning. The model

Table 9: **VITA hyperparameters**.

| Horizons & Observation | |
|---|---|
| Observation horizon | `1` |
| Prediction horizon | `16` |
| Action horizon | `8` |
| Observer backbone | `resnet18` |
| Observer tokenize | `false` |
| **VITA core (latent & losses)** | |
| Latent dimension ($D_{\text{latent}}$) | `512` |
| Decode flow latents | `true` |
| Consistency weight | `1.0` |
| Encoder contrastive weight | `1e-4` |
| Flow contrastive weight | `0.0` |
| Latent noise std | `0.0` |
| **Flow matcher / ODE** | |
| Flow Matcher | `OT-CFM` |
| $\sigma$ | `0.0` |
| # sampling steps (Euler) | `6` |
| **Action AE** | |
| AE recon loss type | `l1` |
| Encoder recon weight | `0.5` |
| Flow recon (FLD) weight | `0.5` |
| Use variational (VAE) | `false` |
| KL weight (if variational) | `1e-6` |
| Freeze encoder / decoder | `false / false` |
| Pretrained path | `None` |
| **AE network (encoder/decoder)** | |
| Encoder type / hidden dim | `cnn / 512` |
| Decoder type / hidden dim | `simple / 512` |
| Latent dim (AE) | `512` |
| Num heads / MLP ratio | `8 / 4` |
| Dropout | `0.0` |
| Num layers | `4` |

is trained for 100 timesteps using a cosine beta schedule. During inference, trajectories are generated using a 10-step DDPM sampler. All hyperparameters and loss weights are listed in Table 11.

## G.4 ACTION CHUNKING TRANSFORMER

We follow the ACT (Zhao et al., 2023) implementation in LeRobot (Cadene et al., 2024). ACT is a conditional variational autoencoder (cVAE) that generates action chunks conditioned on vision. Its architecture consists of a vision encoder for processing observations and a transformer-based decoder that models the distribution of future actions conditioned on the visual input. A complete list of hyperparameters and loss weights is provided in Table 12.

Table 10: **Flow Matching (FM) Policy hyperparameters**.

| Horizons & Observation | |
|---|---|
| Observation horizon | `1` |
| Prediction horizon | `16` |
| Action horizon | `8` |
| Observer backbone | `resnet18` |
| Observer tokenize | `false` |
| **Flow matcher / ODE** | |
| Flow Matcher | `Target` |
| $\sigma$ | `0.0` |
| # sampling steps | `6` |
| **Flow network architecture** | |
| Backbone | `flow_transformer` |
| Conditioning | `adaln` (*options:* `adaln`, `cross`, `cross_adaln`) |
| Hidden dim | `512` |
| Num layers | `4` |
| Num heads | `8` |
| MLP ratio | `4` |
| Dropout | `0.1` |

Table 11: **Diffusion policy hyperparameters**.

| Horizons & Observation | |
|---|---|
| Observation horizon | `1` |
| Prediction horizon | `16` |
| Action horizon | `8` |
| Observer backbone | `resnet18` |
| Observer tokenize | `false` |
| Mask loss for padding | `false` |
| **Diffusion scheduler** | |
| Type | `DDPM` |
| Training timesteps | `100` |
| Beta schedule | `squaredcos_cap_v2` |
| Beta start / end | `1e-4` / `2e-2` |
| Prediction type | `epsilon` |
| Clip sample / range | `true` / `1.0` |
| **U-Net architecture** | |
| Down dims | `[512, 1024, 2048]` or `[256,512,1024]` |
| Kernel size | `5` |
| Group norm groups | `8` |
| Diffusion step embed dim | `128` |
| FiLM scale modulation | `true` |
| **Optimization** | |
| Adam LR / backbone scale | `1e-4` / `0.1` |
| Adam betas / eps | `(0.95, 0.999)` / `1e-8` |
| Weight decay | `1e-6` |
| Scheduler | `cosine`, warmup 500 steps |
| **Training & inference** | |
| Total training steps | `200000` |
| Inference steps | DDPM `10` |

Table 12: Action Chunking Transformer (ACT): hyperparameters.

| Sequence horizons | |
| --- | --- |
| Action horizon | 8 |
| Prediction horizon | 16 |
| Observation horizon | 1 |

| Observer / backbone | |
| --- | --- |
| Image encoder | ResNet-18 |
| Tokenize | false |

| Transformer (policy head) | |
| --- | --- |
| Pre-norm | false |
| Model dimension $d_{\mathrm{model}}$ | 512 |
| Attention heads | 8 |
| Feedforward dim | 3200 |
| FFN activation | ReLU |
| Encoder layers | 4 |
| Decoder layers | 1 |
| Dropout | 0.1 |

| Latent / VAE block | |
| --- | --- |
| Use VAE | true |
| Latent dim | 32 |
| VAE encoder layers | 4 |
| ACT KL weight | 10.0 |

| Optimization | |
| --- | --- |
| Optimizer | Adam |
| Learning rate | $1 \times 10^{-5}$ |
| Betas | (0.9, 0.999) |
| $\epsilon$ | $1 \times 10^{-8}$ |
| Weight decay | $1 \times 10^{-4}$ |
| Backbone LR scale | 0.1 |

| LR scheduler & validation | |
| --- | --- |
| Scheduler | Cosine |
| Warmup steps | 2000 |
| Online validation frequency | 2000 steps |

