# OpenReview forum: "VITA: Vision-to-Action Flow Matching Policy"
_ICLR.cc/2026/Conference — ICLR 2026 Poster_

### Official Review · Reviewer_2nF7 · 2025-10-31

**Soundness:** 2
**Presentation:** 3
**Contribution:** 2
**Rating:** 6
**Confidence:** 3

**Summary:**

This paper introduces Vision-to-Action Flow Matching Policy (VITA), which maps visual representations directly to latent actions using flow matching. Their approach is noise-free and conditioning-free, proposed to reduce time and space complexity. They use an action autoencoder to up-sample action representations to have the same high dimensionality as visual representations. Their approach is simple and relatively novel, achieving state-of-the-art results while being faster.

**Strengths:**

1. This paper's approach is simple and relatively novel in performing flow matching directly from visual representations to action latents.
2. VITA matches state-of-the-art generative policies
3. VITA has a faster inference speed relative to other methods, in part due to its simplicity.
4. The method diagrams they use are simple and understandable (Figures 1 and 2). Their approach is well-motivated and their ablation of FLD and FLC is informative.

**Weaknesses:**

1. The evaluations seem relatively sparse, both in terms of the number of baselines compared and the tasks. For example, how well does FM (or one of the other policies) do on the real-world tasks?
2. The paper alluded to VITA having lower memory footprint than other methods, especially those that use visual conditioning. Can you quantify/estimate this comparison?
3. (minor) Citation formatting issues. For example, many cited papers lack parentheses in the first paragraph of the introduction.

**Questions:**

1. I think the paper would be improved by a little more elaboration on the advantages of being noise-free and conditioning-free, and why others have not got this to work successfully in the past.
2. If VITA used a Transformer/DiT instead of an MLP at roughly the same number of parameters, what would be its latency? (referencing Table 3)
3. Does Table 3 latency include the visual conditioning time?

---

> ### Author Response · Authors · 2025-11-21
> **Responses to Reviewer 2nF7**
>
> We thank the reviewer for acknowledging the simplicity, novelty, and performance of our work. We appreciate the constructive feedback and have added new experiments and revisions accordingly.
>
> ---
>
> ### **1. Adding Baseline Performance for Real-World Tasks**
>
> Thank you for the suggestion. Our ALOHA setup has changed since the submission, and we are currently configuring the new robot arm and re-collecting human demonstrations. Real-robot experiments are time-consuming, but we will report the new results as soon as training and evaluation are completed.
>
> **Updated on Nov 26:** based on the reviewer's constructive comment, we have extended real-world experiments to 5 challeging tasks covering both single-arm and bimanual manipulation, and included comparison to state-of-the-art policy baselines (see the follow-up comment for details). Additionally, we included long-horizon, precision-demanding settings, unseen objects, online perturbations, for more challenging evaluation.
>
> ---
>
> ### **2. Comparing Memory Usage of Baselines and VITA**
>
> We appreciate the reviewer’s suggestion and have added comprehensive memory metrics and detailed discussions in Table 1 and Appendix B.7. The results show that VITA’s flow matching achieves better memory efficiency.
>
> * **VITA (MLP)** has the lowest memory footprint at **333.86 MB**, which is **approximately 19% lower** than FM using the same 4-layer MLP architecture and the best-performing FM variant (Transformer + AdaLN).
> * **VITA (Transformer)** uses **377.55 MB**, outperforming **FM (Transformer + Cross-Attn)** at **529.16 MB**.
>
> ---
>
> ### **3. Citation Formats**
>
> We thank the reviewer for catching the formatting issue. It has been corrected in the revised version.
>
> ---
>
> ### **4. Advantages and Why Prior Approaches Did Not Work**
>
> Thank you for this constructiev feedback. We have improved the clarity of the advantages of our noise-free and conditioning-free formulation (Section 4.1, Introduction, Section 5.2) and expanded our discussion of the challenges in learning vision-to-action flows (Abstract, Introduction, Appendix B.1–B.2). Key points are summarized below.
>
> Conventional FM policies generate samples by starting from random noise and progressively denoising them into the target modality. This requires injecting visual information at every denoising step through conditioning modules, introducing substantial time and memory overhead. For example, cross-attention, a common visual conditioning mechanism, has quadratic time and space complexity. VITA directly flows from visual representations to latent actions. Because the source distribution is visually grounded, VITA eliminates all extra conditioning modules, simplifying the architecture and improving efficiency.
>
> Learning vision-to-action flows is challenging due to:
>
> 1. Dimensionality misalignment between visual and action spaces (Abstract, Appendix B.1).
> 2. The difficulty of learning reliable latent action targets for flow matching (Appendix B.2).
> 3. Latent-space collapse when learning the flow jointly with the evolving latent space (Section 4.3).
>
> These challenges motivated our to introduce the structured latent action space (Appendix B.1) and devising flow latent decoding (FLD, Section 4.4).
>
> ---
>
> ### **5. VITA Using Transformers**
>
> We appreciate the reviewer’s question. We added a new experiment replacing VITA’s MLP with a transformer architecture (31.80M parameters, matching transformer-based FM). The inference latency is **approximately 0.25 ms per chunk**. Notably, VITA is about 2× faster than transformer-based FM (**~0.51 ms per chunk**). Detailed metrics are reported in Table 1 and Section 5.2.1.
>
> ---
>
> ### **6. Whether Latency Metrics Include Visual Conditioning**
>
> Yes. The efficiency metrics in Table 1 report the full inference latency, including visual encoding and all computation required to generate one action chunk from an observation.
>
> ---
>
> **We appreciate the reviewer’s constructive suggestions and hope these clarifications address the raised concerns. We are happy to provide any additional details if needed.**

---

> > ### Comment · Reviewer_2nF7 · 2025-11-23
> >
> > I find the paper's contribution of direct flow matching from visual latent to action latent to be conceptually elegant, and their contribution to improving simplicity and efficiency is really important. I find the validation of the method (e.g., results) to be somewhat sparse still. For instance, one question in my mind is whether this simple VITA approach will work on more challenging benchmarks and more real-world robotic tasks in various setups.
> >
> > But unfortunately, I am not very familiar with the robotics space, so I cannot accurately judge the strength of the results or whether they should be doing more experiments, and I also am unable to recommend additional benchmarks. Overall, the paper seems very interesting and might be improved by substantiating it to make their results more convincing in future iterations.

---

> ### Author Response · Authors · 2025-11-27
> **Response to Reviewer 2nF7's Comment**
>
> We thank the reviewer for recognizing the conceptual elegance of the proposed method, and the importance of simplicity and efficiency in policy learning. We also appreciate the reviewer’s suggestion to further strengthen the empirical validation!
>
> Although our original AV-ALOHA setup is no longer accessible, we have set up a new robot arm, recollected human demonstrations, and re-trained VITA and all baseline policies on **three new challenging real-world tasks** according to the reviewer's suggestion. We have revised the paper with these new results and analyses (Section 5; Table 3 with three real-world tasks and all baselines; Table 2 with a new simulation task; and Appendix E with additional challenging scenarios).
>
> 1. **Expanded evaluation to 9 simulation and 5 real-world tasks.**
>    We extended our evaluation to 9 simulation tasks and 5 real-world robotic tasks. The 3 new real-world tasks we added are challenging due to high randomness or precision requirements.
>
> We included all baselines (FM, DP, ACT). **VITA consistently outperforms or matches state-of-the-art policies in online success rates (Table 3)**.
>
> Furthermore, Figure 15 shows that **VITA converges significantly faster with lower action mean-squared error during training** than other state-of-the-art methods, while ACT plateaus early and DP converges much more slowly.
>
> 2. **Clarification of Benchmarks**
> In recent robotics policy learning works:
>    * DP [1] uses 8 simulation and 4 real-world tasks (single-arm tasks; e.g., PushT, Robomimic, mug flipping)
>    * FM [2] uses 4 simulation and 10 real-world tasks (single-arm tasks; e.g., Robomimic, hanging towels)
>    * ACT [3] uses 6 real-world tasks (bimanual tasks; e.g., cube transfer, slot battery)
>
> VITA follows this standard practice, selecting a number of simulation and real-world tasks. Thanks to the reviewer's suggestion, we make our validation more comprehensive, using 9 simulation and 5 real-world tasks, **covering both single-arm and bimanual tasks**.
>
> Our tasks include these commonly used in existing work (PushT, Robomimic tasks, cube transfer, slot insertion, etc.), with some tasks very difficult for being precision-demanding long-horizon, etc. For example, 1) in a new multi-stage task StoreDrawer, the policy must store an object with random shapes/colors at random positions into a small drawer and close the drawer; 2) our AV-ALOHA tasks also introduce higher action dimensionality with an active vision arm and multi-stage long-horizon structures. in HiddenPick, the policy has to open a box, look inside, and perform pick-and-place (Figure 18).
>
> 3. **Extended Challenges.**
>
>    Following the reviewer’s recommendation, we added experiments evaluating VITA under more challenging real-world conditions:
>
>    * **Online perturbations** during policy rollout (Appendix E; and supplementary video *online_perturbation*)
>    * **Unseen object geometries** for picking and placing (Appendix E; and supplementary video *unseen_objects*)
>
> VITA successfully handles these difficult conditions using limited (50-100) demonstrations.
>
> Again, we sincerely thank the reviewer for the thoughtful comments, which motivated us to significantly improve the breadth and depth of our experimental validation.
>
> [1] Chi, Cheng, et al. "Diffusion policy: Visuomotor policy learning via action diffusion."
>
> [2] Zhang, Fan, and Michael Gienger. "Affordance-based robot manipulation with flow matching."
>
> [3] Zhao, Tony Z., et al. "Learning fine-grained bimanual manipulation with low-cost hardware."

---

> > ### Comment · Reviewer_2nF7 · 2025-11-27
> >
> > I raised my score. The authors addressed my concerns by providing more comprehensive evaluations. Overall, I find their idea conceptually elegant, and their contribution to improving simplicity and efficiency is important.

---

> ### Author Response · Authors · 2025-11-28
> **Response to Reviewer 2nF7’s Follow-Up Comment**
>
> We're delighted to hear that our responses and revised draft have addressed your concerns, and the score was raised to 8.
>
> Thank you for the thoughtful and constructive suggestions on **1)** strengthing our experiments by including more challenging real-world tasks, baselines, and additional experimental setups, **2)** demonstrating VITA's scability to architecture beyond MLPs, by adding experiments of VITA with a transformer architecture, showing VITA retains its efficiency advantages and strong performance, **3)** solidifying the efficiency claims by reporting detailed memory footprints; **4)** further clarifying the advantages of the vision-to-action flow by expanding the discussions.
>
> Your feedback substantially strengthened both the completeness and the clarity of the paper, and we truly appreciate your insight and time!

---

### Official Review · Reviewer_cek7 · 2025-11-01

**Soundness:** 3
**Presentation:** 3
**Contribution:** 3
**Rating:** 4
**Confidence:** 3

**Summary:**

Paper introduces a noise-free, conditioning-free framework that directly maps visual inputs to latent actions using flow matching, eliminating complex denoising and conditioning steps. By combining an action autoencoder with a flow latent decoding mechanism, VITA bridges vision–action dimensional gaps and stabilizes training. Implemented with simple MLPs, it achieves state-of-the-art performance and 1.5×–2.3× faster inference on simulated and real-world robotic tasks compared to conventional diffusion and flow-based policies.

**Strengths:**

- Paper proposes novel solution to the timely and important problem, which seems to be quite significant and useful to the researchers and practitioners.

- Paper itself is cleanly written, explaining the motivation behind the method ingredients. All claims are supported by evidence and additional ablations are provided to further support the proposed contributions. VITA performs on par with the baselines, while being much faster on inference, which is essential for real world applications. Paper provides sufficient information for reproducibility, including hyperparameters.

- Overall, I do not have major concerns regarding the proposed method and think that results are significant.

**Weaknesses:**

I think the paper has two weaknesses.

Firstly, it seems to me that the motivation is not sufficiently explained. After all, why is it important to flow directly from images into actions, rather than from noise with visual conditioning? Why does it introduces complexity? Where does the additional overhead come from? Given that this is not analysed further in the paper, I believe that a simple citation (e.g. line 55) of prior work is insufficient. I advise the authors to elaborate on their reasoning in greater detail.

Second, It seems to me that the introduction of MLP-backbone is rather sudden and not particularly justified, in the sense that it is an additional confounder to the main method, which invalidates a fair comparison of approaches specifically for predicting actions. And there are no ablations on that. What if the main speed up (or even performance gains) come from the simpler backbone? How would baselines perform with similar backbones? If baselines will fail but VITA is not, what is the main reason?

I can not answer this questions from the current paper results and this is critical.

**Questions:**

1. Can authors provide std or (better) confidence intervals for Table 3? It should be possible, given that you used more than one random seed.
2. Why set of baselines differ between experiments (Table 1 & Table 3)? Won't the results differ from other conditioning methods, e.g. AdaLN, cross-attention, FiLM?
3. I think this line needs clarifications: “we use vector representations for both vision and action, further reducing time and space overhead”. To my knowledge, most existing methods represents observations as vectors… What is the difference specifically and why it reduces time overhead?
4. Can baselines be trained with MLP backbones? Would they be faster or still slower than the proposed VITA?

Misc:

typo on 81 line “collapse of targe latent action”

---

> ### Author Response · Authors · 2025-11-21
> **Responses to Reviewer cek7**
>
> We thank the reviewer for acknowledging the importance of the research problem, and the performance and significance of the results.  We appreciate the reviewer’s constructive comments that help us improve the clarity! We conducted multiple new experiments, and revised the paper according to the suggestions.
>
> ### **W1 Why Vision-to-Action Flow is Important**
>
> We thank the reviewer for the constructive comment! We revised Introduction, Section 5.2.1, Section 4.1 (changes highlighted) to clarify why flowing from vision to action is important. We added detailed discussions/experiments results of why extra conditioning modules for flow matching introduce more overhead.
>
> 1. Section 5.2.1 (Table 1) and Appendix B.7 (Table 5) now include more detailed time and memory usage to quantify the overhead introduced by extra conditioning modules.
> 2. Conventional FM process requires repeatedly injecting visual information at each denoising step through additional conditioning modules, resulting in substantial time and space overhead. E.g., cross-attention is often used for visual conditioning, and incurs quadratic time and space complexity. Minimizing complexity is essential for real-time robot control. For example, Helix controls at 200 Hz, necessitating low-latency inference. The primary objective of VITA is to overcome the inefficiencies inherent to conditioning in conventional flow matching. VITA directly flows from visual representations to latent actions. Since the source of the flow is visually grounded rather than random noise, VITA eliminates the extra conditioning modules and reduces complexity.
>
> ### **W2 Ablation on MLP Architectures**
>
> We appreciate the insightful comment, and conducted more ablations. Table 1 compares the efficiency of FM (MLP) vs. VITA (MLP), and FM (Transformer) vs. VITA (Transformer). VITA achieves 1.5× and 2× speedup, and 18.6% and 28.7% reduction in memory, when using MLP and transformer, respectively. Section 5.2.1 discusses that the efficiency gains are not derived from specific architectures, but from the removal of conditioning.
>
> We added experiments to evaluate the performance of FM (MLP) (Appendix B.6.1), where FM performs poorly because MLP fails to process high-dimensional noisy actions and fuse in visual conditions. In contrast, VITA (MLP) surpasses or matches transformer-based FM, because VITA simplifies the flow matching network to vector-to-vector mapping without extra conditioning and effectively bridges the vision and actions.
>
> ### **Q1 Success Rates with STD**
>
> Thanks for the suggestion. We have provided the std over 3 random seeds for all the tasks in Table 2.
>
> ### **Q2 Why Baselines Differ in Table 1 & 2**
>
> We added clarifications to the paper  Section 5.2.2 according to the reviewer’s comment. Table 1 included multiple widely used architectures and conditioning modules for a comprehensive efficiency comparison. In the performance comparison, we report transformer-based FM using AdaLN among 3 different FM setups, because it overall yields the best performance. MLP-based FM yields poor SRs (Appendix B.6.1), and transformer-based FM using cross-attention yields similar SRs compared to AdaLN but is much slower to train.
>
>
> ### **Q3 Vector Representation for Overhead Reduction**
>
> We have revised the paper to improve the clarity (Introduction, Appendix B.6). While existing methods may represent observations as vectors, they differ in how these vectors are used. In conventional FM, visual observations are repeatedly injected into the flow at every denoising step via conditioning modules (e.g., cross-attention), incurring time/space overhead. In contrast, VITA treats the visual vector as the source of the flow, and therefore, eliminates conditioning. The flow operates over compact vector representations of from vision source to action target, simplifying the flow matching network to vector-to-vector mapping without extra conditioning, leading to lower memory and faster inference, as evidenced by our detailed latency and memory benchmarks (Table 1). Additionally, we added detailed discussions (Section 5.2.1) and conducted experiments (Appendix B.6) to show that the efficiency gains of VITA are not limited to vector-based representations or MLP. We implemented VITA using grid-based features and transformers, and show that VITA still speeds up 2x and reduces memory use by 28.7\% (Table 1).
>
> ### **Q4 Latency of Baseline using MLP**
>
> Resonses also provided in W2. VITA is still 19% faster than FM implemented using MLP (Table 1).

---

> > ### Comment · Reviewer_cek7 · 2025-11-27
> >
> > I thank the authors for their thoughtful rebuttal. All of my concerns have been addressed, and I have no further questions. I raised by score.

---

> ### Author Response · Authors · 2025-11-28
> **Follow-Up Response to Reviewer cek7**
>
> We're very glad that our revisions and clarifications addressed your concerns, and the score was raised to 8.
>
> We thank the reviewer for the insightful suggestions on **1)** strengthening the presentation of our motivation by providing more details and reasoning why the removal of conditioning is beneficial **2)**  ablating the architecture choice, showing the VITA achieves efficency gains and maintains strong performance across different architectures with expanded experiments; **3)** removing the confounder of visual representation choices by adding experiments for both vector-based and grid-based representations to highlight VITA's scalability **4)** reporting std in success rates, and clarifying the selection of baselines in Table 1 and 2 and the reason of using transformer-based FM to report SRs due to its highest performance.
>
> Your constructive feedback substantially strengthened our paper, and we sincerely appreciate the time and efforts you dedicated to the review.

---

### Official Review · Reviewer_od34 · 2025-11-01

**Soundness:** 2
**Presentation:** 2
**Contribution:** 2
**Rating:** 4
**Confidence:** 4

**Summary:**

This paper introduces a framework that maps visual observations directly into latent action spaces through a flow-matching approach, avoiding traditional denoising or conditioning pipelines. By integrating an action autoencoder with a latent flow decoder, the proposed method (VITA) aligns vision and action representations and improves training stability. Implemented with lightweight MLP networks, it achieves comparable results to diffusion-based and flow-based policies while delivering faster inference across simulated and real-world control benchmarks.

**Strengths:**

The idea of simplifying visuomotor policy learning by removing conditioning mechanisms is potentially interesting. The paper is generally clear in presentation, and experiments are competently executed. The implementation details are sufficiently documented, and the inclusion of ablation studies is appreciated.

**Weaknesses:**

The main conceptual motivation is underdeveloped. The authors state that removing conditioning simplifies the process, yet the argument remains superficial. It is not clear why direct flow from vision to actions should be advantageous, or what specific drawbacks the previous conditioning-based methods introduce. Without a stronger analysis, the contribution feels somewhat incremental.

The use of MLP backbones further complicates interpretation of results. Introducing such a lightweight architecture changes the comparison dynamics, and without explicit ablations, it’s unclear whether performance and efficiency gains originate from the proposed framework or simply from the reduced model complexity. This limits the credibility of the reported advantages.

Overall, while the approach is clean and runs efficiently, it does not convincingly establish the necessity or distinctiveness of the proposed design choices.

**Questions:**

(1) Have the authors evaluated whether the choice of an MLP backbone influences the observed improvements? Specifically, if the same MLP architecture were used for the baseline methods, would their inference times and performance still trail behind VITA? How much of the reported speed-up is due to the backbone rather than the proposed flow-matching formulation itself?

(2) Have the authors evaluated whether the choice of an MLP backbone influences the observed improvements? Specifically, if the same MLP architecture were used for the baseline methods, would their inference times and performance still trail behind VITA? How much of the reported speed-up is due to the backbone rather than the proposed flow-matching formulation itself?

(3) Why do the sets of baselines differ between experiments (e.g., Table 1 vs. Table 3)? What was the rationale for omitting certain conditioning-based approaches such as AdaLN, FiLM, which seem directly relevant to the claimed simplifications? Are the results expected to hold under a unified baseline setup, or were baselines adjusted for computational or implementation reasons?

---

> ### Author Response · Authors · 2025-11-21
> **Responses to Reviewer od34**
>
> We thank the reviewer for constructive questions and feedback! We have added several new experiments, improved the presentation of our motivation, and added several clarifications throughout according to the comments.
>
> ### **W1 Clarity of Motivation & Advantages**
>
> We appreciate the suggestion to improve the clarity of our paper. We have expanded the discussions of our motivation in the Introduction, Section 4.1, and Section 5.2.1 to include a more comprehensive explanation of why flowing from vision to actions is advantageous.
>
> Conventional flow matching (FM) policies generate samples by starting with random noise and progressively “denoising” them into the target modality. This process requires **repeatedly** injecting visual information at each denoising step through additional conditioning modules, resulting in substantial time and space overhead. For example, cross-attention is often used for visual conditioning, but it incurs quadratic time and space complexity.
>
> Minimizing complexity is essential for real-time robot control. For example, Helix controls at 200 Hz, which necessitates low-latency inference. The primary objective of this paper is to overcome the inefficiencies inherent to conditioning mechanisms in conventional flow matching methods. To this end, we propose VITA, a noise-free policy learning framework that directly flows from visual representations to latent actions. **Since the source of the flow is visually grounded, VITA eliminates the need of extra conditioning modules and therefore reduces complexity.**
>
> ---
>
> ### **W2 & Q1 Ablations on Architecture Choices**
>
> We thank the reviewer for the proposed ablation! We have conducted several new ablation experiments for a more comprehensive analysis and revised the paper to show that the efficiency and performance gains are enabled by VITA learning rather than specific architectural design choices (Section 5.2.1).
>
> Specifically,  we implemented MLP-based FM to compare FM and VITA when both are using lightweight MLPs, Table 1 shows that VITA is still **1.3x faster and consumes 18% less memory**. Furthermore, we implemented transformer-based VITA in comparison with transformer-based FM. When both VITA and FM use complex transformer architectures, VITA retains the efficiency gains, achieving a **2× inference speedup** (detailed in Section 5.2.1) and a **28.7% reduction in memory usage**. The above experiments ablate the architecture choice and show that efficiency gains of VITA are not limited to specific architectures.
>
> Furthermore, Appendix B.6.1 and Figure 11 show that when VITA and FM are both using simple  MLP architectures, FM fails to learn reasonable policies and action MSEs plateau quickly because the MLP fails to process noisy action chunks while fusing in the visual conditions. On the other hand, MLP-based VITA surpasses or matches the best-performing FM variant (transformer-based) on all the tasks, showing the performance gains are origingated from the the proposed method rather than specific architecture choices.
>
> ### **Q2 Baseline Selection in Table 1 & 2**
>
> We have added clarifications to the Section 5.2.2 according to the reviewer’s comment regarding why Table 1 and Table seem to have different baselines. Table 1 includes multiple widely used architectures and conditioning modules for a comprehensive efficiency comparison. In the performance comparison of Table 2, we report transformer-based FM using AdaLN among three different FM setups because it yields the best performance overall. MLP-based FM yields poor SRs (Appendix B.6.1), and transformer-based FM using cross-attention achieves similar SRs to AdaLN but is much slower to train.

---

### Official Review · Reviewer_7Vvz · 2025-11-01

**Soundness:** 2
**Presentation:** 2
**Contribution:** 1
**Rating:** 2
**Confidence:** 4

**Summary:**

This paper is about VITA, a noise-free and conditioning-free policy learning framework that attempts to directly map visual representations to latent actions using flow matching. The authors introduce an action autoencoder to align the dimensionalities of the vision and action spaces and propose FLD to stabilize the joint training by backpropagating the reconstruction loss through the ODE solving steps.

**Strengths:**

- I like the core idea of using the visual latent space as the source distribution for the flow matching process. Removing the explicit conditioning mechanisms (like cross-attention or FiLM) that are standard in diffusion/flow policies simplifies the architecture and naturally leads to faster inference.

- The resulting architecture is lightweight. It is compelling that an MLP-only network for the flow matching and decoding components (following the ResNet encoder) can handle complex bimanual tasks like those in the ALOHA suite.

**Weaknesses:**

- I find the claim of being "conditioning-free" (L014) slightly misleading. While explicit conditioning modules are removed, the flow is inherently conditioned on the visual input because the visual latent is the source distribution (z0). The velocity field must learn the transport from this specific starting point. This feels more like implicit conditioning via the ODE initial state rather than a fundamental removal of conditioning.

- The approach seems heavily constrained by the architecture. The MLP-only implementation relies on a global average-pooled ResNet-18 feature vector. This severely compresses spatial information and likely bottlenecks the policy's ability to handle tasks requiring fine-grained spatial understanding. It is unclear how this approach would scale if we switched to representations that retain spatial information (e.g., spatial tokens), which often require more complex backbones like U-Nets or Transformers, potentially nullifying the efficiency gains.

- The experimental results are mixed and the baseline comparisons are questionable. VITA underperforms conventional FM on 'Slot Insertion' (0.76 vs 0.83) and 'PourTest Tube' (0.79 vs 0.84). Furthermore, the baseline performance is suspiciously low in some cases (e.g., DP achieving 0.00 on ThreadNeedle), even though the authors mention increasing training steps for DP and ACT significantly. This makes it difficult to gauge the true strength of VITA.

- The necessity of FLD introduces a significant hidden cost. FLD requires backpropagation through the ODE solver steps during training. This substantially increases the computational graph complexity and memory requirements during training. The paper focuses heavily on inference speed, but the practical training overhead of FLD is not discussed or quantified.

- The efficiency gains presented in Table 3 mix architectures and conditioning methods. VITA (MLP) is compared against FM implemented with DiT or U-Net. A fairer comparison would be conventional FM using the same MLP architecture and a simple conditioning mechanism to isolate the benefit of the noise-free approach from the architecture choice.

**Questions:**

- Regarding the tasks where VITA underperforms FM, what is the hypothesis for this performance drop? Does the reliance on the globally pooled visual feature bottleneck performance on these precision tasks?

- What is the actual overhead in training time and memory usage when enabling FLD?

- Why did Diffusion Policy fail so completely on ThreadNeedle and HookPackage despite extended training? Were the hyperparameters for the baselines thoroughly tuned?

- In Section 4.3, paper mentions that a pre-trained and frozen action AE is ineffective. Could you provide the empirical results for this ablation?

- Have you experimented with representations that retain more spatial information instead of the global average-pooled vector? If so, would the MLP architecture still be sufficient?

- How does VITA handle multimodal action distributions? In conventional diffusion/FM, stochasticity comes from the initial noise sampling. In VITA, the process seems largely deterministic once the image latent z0 is fixed. How is multimodality captured?

---

> ### Author Response · Authors · 2025-11-21
> **Responses to Reviewer 7Vvz**
>
> We thank the reviewer for appreciating the core idea and lightweight architecture of VITA, and for the insightful questions and feedback. We have incorporated new experiments, reorganized the presentation of our evaluations, and made numerous clarifications in the revised submission.
>
> ### W1 The Conditioning-Free Claim
> We clarify that the VITA flow matching process is conditioning-free because it learns a velocity field v(·), not a conditional velocity field v(· | O), as in conventional FM policies. We added clarification to Section 4.1 (changes are highlighted) to illustrate the conditioning-free flow matching per the comment.
>
> ### W3 & Q3 DP Failures & Underperformance on Two Tasks
>
> We have investigated and addressed the DP failures after finetuning. Table 2 reflects the changes. We added further discussions for the DP failures in Appendix B.8.2 and below.  We found that min-max instead of mean-std normalization [1] effectively fixes the failures of DDPM on our ALOHA datasets. The remaining datasets with proper normalization were not affected. We thank the reviewer for the catch, which solidifies our DP baselines and deepens our understanding.   Appendix B.8.2 shows that DP underperforms mainly because some tasks are extremely precision-demanding. For ThreadNeedle, the policy must complete all five subtasks (reward=5.0) to succeed. Tiny pose errors lead to failures. Although DP achieves ~4.0 rewards in training, it fails the strict success criteria with ~40% SR, while VITA/FM reaches 4.7+ rewards and ~90% SRs. Additionally, we added **Figure 15**, which compares the action mean-squared-error curves during training. The results show that VITA converges significantly faster than DP and ACT; ACT plateaus at higher errors, while DP learns more slowly, further demonstrating VITA’s advantages in precision and training efficiency. Regarding VITA underperforming FM on two tasks, we’d like to emphasize that VITA matches or outperforms FM on 7 of 9 tasks while being 150% faster than FM. Introduction and Sec 5.2 have emphasized that the primary objective of this paper is to improve time and space efficiency rather than success rates.
>
> ### W4 & Q2 Training Overhead of FLD
>
> We thank the reviewer for proposing the addition of FLD overhead! Table 5 and Appendix B.7.1 discussed FLD and its efficiency metrics. The time and space overhead introduced by FLD in training is 9.3% in time and 4.4% in space, while VITA still consumes less memory, and remains comparable in training time, compared to other baselines.  FLD overhead remains minimal, since the ODE is solved by calling the lightweight VITA network with only 6 discretization steps. We clarify that  the 1.5x–2x inference speedup, which is crucial for real-time deployment,  justifies the modest training overhead, because faster inference is critical for real-time deployment.
>
> ### Q1 Is the pooled feature bottleneck?
>
> It’s sensible to start with the hypothesis, but after further investigation, we clarify that both VITA and FM use pooled features. VITA is performing strongly on ThreadNeedle, which is one of the most challenging tasks that requires high precision, which  also contradicts the hypothesis that pooled features pose a bottleneck.
>
> ### W5 FM using MLP
>
> We thank the reviewer for proposing the ablation! New experiments using the same MLP architecture for FM are conducted, with explanations in Appendix B.6.1. As shown in Fig. 11, MLP-based FM failed to learn reasonable policies due to limited capacity, highlighting that VITA’s benefits stem from its vision-to-action flow formulation rather than architectural choices.
>
> ### W2 & Q5 VITA with spatial tokens transformers
>
> We conducted extensive experiments to evaluate VITA’s scalability with spatial tokens and complex architectures like transformers according to the reviewer's suggestion. Section 5.1 and Section 5.2.1 now include detailed discussions on both vector- and grid-based features, showing that VITA maintains efficiency while achieving comparable performance. Table 1, Appendix B.6.2, and Appendix B.7 provide comprehensive metrics confirming that VITA’s efficiency gains (2x speed up and 28.7% less memory) persist even with more complex backbones.
>
> ### Q4 Frozen AE ablation
>
> We added the frozen action AE ablation to the paper (Appendix Section B.2 and Figure 8) according to the reviewer's suggestion.
>
> ### Q6 Stochastic Sampling of VITA
>
> For the multi-modality question, we explored several ways to introduce stochasticity into VITA, including using SDE instead of ODE, applying a variational vision encoder, and injecting covariance into visual latents. These variants are discussed in Appendix B.5 and B.8.1. As noted, we found that adding stochasticity can degrade performance on precision-sensitive tasks, as it introduces noise into the visual latents, reducing the fidelity of the visual and action details.
>
> [1] Chi, Cheng, et al. "Diffusion Policy: Visuomotor Policy Learning via Action Diffusion."

---

### Author Response · Authors · 2025-12-03
**Rebuttal Summary**

**To the Area Chair and Reviewers:**
We thank the reviewers for their thoughtful feedback and constructive engagement. After additional experiments and clarifications, **Reviewers cek7 and 2nF7 confirmed that all their concerns were addressed and raised scores to 8**. We appreciate the reviewers for highlighting the “conceptually elegant” nature of VITA, the significance of improving efficiency and simplicity, and the strength of our experiments. We summarize comments and responses on motivation, architecture, experiments, and efficiency. See supplementary `demos.html` for demo videos.

**Clarity of Motivation**
Reviewers cek7, 2nF7, and od34 suggested providing more details about advantages of our noise-free and conditioning-free formulation over conventional flow matching (FM). We clarify that conventional FM policies hinge heavily on repeatedly injecting visual inputs (via cross-attention, AdaLN, or FiLM) at every generation step (Abs, Intro). VITA grounds the flow source directly in the visual representation, thus removing the need for conditioning during denoising. Further experiments quantify efficiency gains (Sec 5.2.1, Tab 1).

**Architecture**
* **Is efficiency due to the MLP backbone? How does FM + MLP perform? (7Vvz, od34, cek7):** We demonstrated that efficiency gains stem from the conditioning-free formulation, not just architectures. We implemented `FM + MLP` and compared it with `VITA + MLP` (App B.6.1). `FM + MLP` fails to learn effective policies, whereas `VITA + MLP` matches or outperforms the best-performing FM variant (transformer) while being **1.3x faster and 18.6% more memory-efficient** than the most efficient FM variant (MLP).
* **Scalability to Other Architectures (2nF7, 7Vvz):** To prove VITA is not architecture-constrained, we evaluated `VITA + Transformer`. It remains **2x faster with 28.7% reduced memory* compared to `FM + Transformer` of similar model size, while maintaining strong performance (Tab 1, App B.6.2).

**Visual Representation**
* **Clarifications on the Choice of Vector or Grid-Based Visual Features (cek7, 7Vvz):** VITA improves efficiency regardless of the representation choice (Sec 5.2.1) supported by experiments (Tab 1).
* **Scalability to Spatial Features (7Vvz):** Added VITA using spatial features showing VITA attains strong performance (App B.6.2) and superior efficiency (Tab 1).
* **Are Pooled Features Performance Bottlenecks due to Compressed Information (7Vvz):** VITA and FM (vector-based settings) in Tab 2 both leverage pooled features and achieve state-of-the-art results on our most challenging precision-demanding tasks like **ThreadNeedle (92% SR, success rate)**, contradicting the hypothesis that pooled features pose a bottleneck. As also suggested by cek7, most existing policies use pooled features.

**Strengthening Tasks & Baselines**
* **Expanded Tasks (2nF7):** We expanded experiments to **9 simulation and 5 real-world tasks**, covering bimanual and single-arm manipulation. Several tasks are challenging for being precision-demanding, long-horizon, and highly randomized. Additionally, VITA handles two new real-world challenges robustly: **online perturbations** and **unseen objects** (App E).
* **DP Under-Performance (7Vvz):** Strengthened DP baseline by fixing a normalization issue (using min-max), and re-ran 3 seeds. We added explanations for the low SRs of DP, which is largely due to the artifact with strict success criteria in multi-stage ALOHA tasks where millimeter errors lead to binary failures, highlighting VITA's superior precision (App B.8.2).

**Training Overhead & Memory Costs**
* **7Vvz suggested Flow Latent Decoding (FLD) Introduces Significant Training Overhead:** We measured training efficiency (Tab 6) and found FLD introduces only modest overheads (**9.3% time, 4.4% space**) because VITA ODE takes only 6 steps using conditioning-free networks. This slight training cost enables effective e2e optimization for VITA to yield substantial inference gains (**1.5x–2x speedup, 18.6%-28.7% memory reduction**), which is **critical for real-time robotic control**.
* **More Efficiency Metrics (2nF7, 7Vvz)**: Detailed training/inference time/memory metrics Tab 1 & Tab 6, showing superior efficiency of VITA.

**More Clarifications**
* **Why VITA is Conditioning-Free (7Vvz):** VITA learns a velocity field $v(z_t, t)$ rather than a conditioned field $v(z_t, t | O)$.
* **How to Introduce Stochasticity and Capture Multimodality (7Vvz):** We explored using SDEs instead of ODEs, variational encoders, or adding covariance to the source (App B.5). We found that less stochasticity achieves superior precision for fine-grained manipulation, as stochasticity can degrade the fidelity of vision and action details.
* **Ablation of Frozen Autoencoder (7Vvz):** Added an ablation of frozen autoencoder (App B.2). The action MSE plateaus early since target latents cannot be fixed during flow matching training, showing the necessity of end-to-end optimization.

---

### Meta-Review · Area_Chair_euPh · 2026-01-04

**Summary:**

Reviewers uniformly acknowledge the paper’s core idea, its simplicity, as well as its novelty to a timely problem. However, several critical concerns converge across reviews regarding the use of MLP backbones, baseline/task selection in experiments, and the need for key clarifications.

**Reviewer Concerns:**

The rebuttal substantively strengthens the paper’s claims by providing more evaluation results and clarifications.

**Reviewer Scores:**

Reviewer cek7 raised their score from 4 to 8, and 2nF7 from 6 to 8, bringing the average to 5.5.
Given that the remaining reviewers (7Vvz and od34) raised concerns largely overlapping with those addressed in the rebuttal (e.g., architecture confounding, baselines and tasks, motivations), and the authors comprehensively resolved these points with new experiments and clarifications, it is reasonable to expect one or both would also increase their scores. The consensus thus shifts clearly toward acceptance.

---

### Decision · Program_Chairs · 2026-01-26

Accept (Poster)